behaviour/evolution

eusocial insects, animal sociality, predation, communication, alarm signals, multimodal signals

**Author for correspondence:**
Heather R. Mattila
e-mail: hmattila@wellesley.edu

†Present address: Champalimaud Research, Champalimaud Centre for the Unknown, Lisboa, Portugal.
‡Present address: Department of Entomology, The Pennsylvania State University, University Park, PA, USA.

# Giant hornet (*Vespa soror*) attacks trigger frenetic antipredator signalling in honeybee (*Apis cerana*) colonies

Heather R. Mattila[1], Hannah G. Kernen[1], Gard W. Otis[2], Lien T. P. Nguyen[3], Hanh D. Pham[4], Olivia M. Knight[2,†] and Ngoc T. Phan[5,‡]

[1]Department of Biological Sciences, Wellesley College, Wellesley, MA, USA
[2]School of Environmental Sciences, University of Guelph, Guelph, Ontario, Canada
[3]Insect Ecology Department, Institute of Ecology and Biological Resources, Vietnam Academy of Science and Technology, Hanoi, Vietnam
[4]Bee Research Centre, National Institute of Animal Sciences, Hanoi, Vietnam
[5]Research Center for Tropical Bees and Beekeeping, Vietnam National University of Agriculture, Hanoi, Vietnam

HRM, 0000-0001-5712-1688

Asian honeybees use an impressive array of strategies to protect nests from hornet attacks, although little is understood about how antipredator signals coordinate defences. We compared vibroacoustic signalling and defensive responses of *Apis cerana* colonies that were attacked by either the group-hunting giant hornet *Vespa soror* or the smaller, solitary-hunting hornet *Vespa velutina*. *Apis cerana* colonies produced hisses, brief stop signals and longer pipes under hornet-free conditions. However, hornet-attack stimuli—and *V. soror* workers in particular—triggered dramatic increases in signalling rates within colonies. Soundscapes were cacophonous when *V. soror* predators were directly outside of nests, in part because of frenetic production of antipredator pipes, a previously undescribed signal. Antipredator pipes share acoustic traits with alarm shrieks, fear screams and panic calls of primates, birds and meerkats. Workers making antipredator pipes exposed their Nasonov gland, suggesting the potential for multimodal alarm signalling that warns nestmates about the presence of dangerous hornets and assembles workers for defence. Concurrent observations of nest entrances showed an increase in worker activities that support effective defences against giant hornets. *Apis cerana* workers flexibly employ a

diverse alarm repertoire in response to attack attributes, mirroring features of sophisticated alarm calling in socially complex vertebrates.

## 1. Introduction

One of the most intriguing features of animal sociality is the evolution of shared signals that convey information and coordinate activity among group members [1–5]. Predation is a major selective pressure for animals that live in conspicuous social groups, and the rich antipredator signalling that it drives can reveal the intricacies of social communication [6,7]. Signal meaning can be revealed by immediate responses to predation threats, both in the production of signals by alarmed individuals and the response of group members to those signals. Furthermore, selection should favour signal diversity in species that are hunted by predators that differ in attack strategy, the degree of danger they pose to prey or prey response [8,9]. Importantly, for social animals that respond collectively to predators, signals organize group-level defences [7,10,11]. Signals produced in response to predators may encode predator type, level of urgency or both [12–15]. These signals may be discrete or graded, meaning they may have distinct features that discriminate them from other signal types or they may vary on a continuum with intermediate forms [9,16–18]. Finally, antipredator signals may be multimodal, which can refine their influence on recipients, aid communication in noisy environments and help group members respond appropriately when attacks come from multiple types of predators [9,19–22].

The emerging picture is that one needs to know an animal species well in order to understand how group members communicate when faced with predatory threats [9,23]. Acoustic monitoring is an excellent way to gain valuable insights into the signals that social groups exchange as they detect predators and coordinate defensive responses, particularly in environments where sound is a well-used modality and visual observation is challenging [24–28]. Honeybees (genus *Apis*) are an important model system for exploring signal use within a social group because of the diversity of 'sounds' that colony members exchange to coordinate their activities [29–33]. Honeybees perceive sounds either as air-particle movements detected by Johnston's organs in their antennae or as substrate-borne vibrations detected by subgenual organs in their legs [34–36]. Thus, bee-produced signals are collectively termed 'vibroacoustic' because they are often transmitted within colonies simultaneously as both airborne sounds and substrate vibrations, and mode of perception is not always clear [32,33].

The perception of airborne sounds by honeybees is presently understood to be limited to the brief pulses (less than 50 ms) made by waggle-dancing workers in several species of *Apis* [37–45]. By contrast, a class of substrate-borne vibrations called 'pipes' are produced by workers in many contexts, including responses to predatory threats [46–52], responses to conditions at food sources [39,40,53–58], during the swarming process [31,59–63] and when queenless [47]. A pipe is produced when a worker vibrates her thorax and presses her body against a substrate to transmit the vibration (reviewed [32,33]), generating a characteristic harmonic structure when visualized in spectrograms [31,48,51,56,64]. Worker-produced pipes were first described a century ago [65] and their production and function have been best studied in the European honeybee, *Apis mellifera*. For instance, a subset of brief pipes called 'stop signals' are produced by workers in *A. mellifera* nests and swarms; in both social contexts, stop signals inhibit waggle dancing by receivers [49,53,54,57,63]. Within nests, they reduce recruitment to perilous food sources [49,50,58], whereas in swarms they suppress recruitment to competing nest sites [63]. In *A. mellifera*, stop signals have mean durations reported as 142–230 ms and fundamental frequencies of 270–540 Hz, and they are often delivered while signalling workers butt their heads against recipients' bodies [39,40,54,56,58,64]. Beyond well-characterized stop signals, features of *A. mellifera* pipes can vary in several ways. Workers often produce pipes that are much longer than stop signals (e.g. greater than 2 s at the extreme) and they can vary how pipes are delivered, for example by pressing their bodies onto other workers or nest surfaces [31,55,56,64]. In swarms, longer pipes trigger preparation for liftoff [31,64], but it is not known how workers respond to long pipes made in nests.

Unlike *A. mellifera*, the use of vibroacoustic signals by other species of honeybees, all of which are endemic to Asia [66,67], is not as well studied. However, because of the strong predation pressure Asian honeybees face [68], most studies of their vibroacoustic signals have focused on alarm signalling. Hornets (genus *Vespa*) are Asian honeybees' most persistent and damaging predators [68–72], and early studies have noted the audible pipes and hisses colonies make when they are attacked by hornets [47,68,70,73–77]. Stop signals have been widely observed in the genus *Apis* [39], but their

function has been studied in *A. cerana* only among the Asian honeybees. *Apis cerana* workers exposed to tethered hornets (alive or dead) adjust the traits of the stop signals they produce in response to attack attributes, and signal recipients are less likely to perform recruitment dances or leave the safety of the nest [51,52]. In *A. florea*, the presence of threatening stimuli near nests induces workers to pipe, which in turn triggers group hissing [48]. Hisses are produced when many workers move their bodies and vibrate their wings synchronously in response to mechanical disturbance or predator attack, including harassment by hornets [48,68,77–80]. Hisses are often produced in series and they may be shorter in duration when hornets are present [48,77,80], but *A. cerana* colonies also hiss when disturbances are not apparent [80]. While the function of hisses is not clear, they are proposed to be aposematic warnings to predators and may also reduce the activity of nestmates to lower their predation risk [11,48,68,77,80]. For both hisses and stop signals, colonies increase signalling rate in the wake of predatory threats [48,51]. Thus, Asian honeybees employ discrete categories of vibroacoustic alarm signals and colonies adjust signal parameters in response to attack attributes. However, much remains to be discovered about how honeybees use vibroacoustic signalling to coordinate antipredator behaviour as they defend their nests.

This study explores the signalling repertoire of *A. cerana* during naturally occurring attacks by two hornet predators that differ in the degree of threat they pose to colonies. At our study site in Vietnam, the deadliest hornet predator that *A. cerana* encounters is *Vespa soror*, a giant hornet that can decimate honeybee colonies through group predation [81,82]. A successful attack starts when a *V. soror* scout recruits nestmates to a prey colony, where together they kill many of the defending honeybees, occupy their nest and harvest undefended brood to feed their larvae. *Vespa soror* is not well studied, but it is morphologically and behaviourally similar to its better-known sister species, the giant hornet *Vespa mandarinia* [70,71,81–87]. By contrast to the two species of giant hornets, *Vespa velutina* is a smaller hornet that hunts solitarily by hawking individual honeybees while hovering in front of nests [72]. In the evolutionary arms race between predator and prey, *A. cerana* has evolved several colony-level defences to fend off hornet attacks. They often aggregate at the nest entrance as a first step [70,88,89], referred to as a 'bee carpet' in *A. mellifera* [90–93]. Once amassed, workers can engulf an individual hornet in a ball of hundreds of bees, simultaneously overheating and asphyxiating it [89,94–96]. *Apis cerana* workers apply materials (i.e. animal faeces in Vietnam, plant material in Japan) around nest entrances to repel giant hornets, a defensive behaviour that is not triggered by smaller hornets [82,97]. Groups of workers also perform coordinated body shaking in response to hornets, a visually intimidating display that deters attackers from approaching the nest [77,98–101].

These sophisticated defences require timely predator detection and swift activation of a defending workforce. Vibroacoustic signals likely play an important role in organizing these responses because they are transmitted quickly between senders and receivers within nests [29,33]. We comprehensively catalogued vibroacoustic signals captured in colony soundscapes as *A. cerana* workers responded to attack by two varyingly dangerous predators: *V. soror*, a giant hornet that launches group attacks on colonies, and *V. velutina*, a smaller hornet that hunts solitarily. Our findings highlight striking differences in the signalling response of *A. cerana* colonies to these two predators. Colony soundscapes showcase the diversity of *A. cerana*'s alarm signalling repertoire, including a novel antipredator pipe made by workers when *V. soror* workers were present at nest entrances. Simultaneously recorded videos of nest entrances demonstrate that changes in colony-level signalling are linked to the arrival of hunting hornets and the initiation of activities by worker bees that support predator-specific nest defences.

# 2. Methods

## 2.1. Study site and colonies

All studies were conducted in three apiaries of *A. cerana* colonies managed by beekeepers in the communes of Ba Trai and Cam Thượng, Ba Vì District, Hanoi Province, Vietnam. Two apiaries were located 1 km from each other and both were approximately 13 km away from the third apiary (GPS coordinates: Apiary 1 = 21.118 N, 105.335 E; Apiary 2 = 21.105 N, 105.335 E; Apiary 3 = 21.168 N, 105.447 E). Apiaries had 136, 55 and 22 colonies, respectively. All colonies were managed in wooden hives that had three frames of comb and a small entrance to limit hornet entry (2–8 cm long and less than 1 cm high). Occasionally, we transferred colonies into two-frame observation hives prior to collecting data (four controls out of 32 replicates; see next section) by installing frames with the most

brood and food from a hive and shaking all bees into the observation hive. Entrances of observation hives were large enough to permit hornet entry, although this did not occur. All fieldwork was conducted from late August through October 2013, when *Vespa* predation on *A. cerana* colonies is anticipated by local beekeepers, and only on days when the weather permitted foraging by bees and hunting by hornets.

## 2.2. Recording colony signalling during hornet-attack scenarios

We recorded signals produced in colonies in response to five hornet-attack scenarios, including naturally occurring attacks by *V. soror* or *V. velutina* workers, exposure to van der Vecht gland (VG) extracts from *V. soror* workers (the source of a marking pheromone that scouts use to target nests, in ether), and two no-attack controls (hornet-free conditions; including an ether-sham treatment). We did not control the access of hornets to colonies, so a recording of a real attack included periods when one or more hornets were in front of a hive, as well as periods in the aftermath of their departure, when no hornets were present. Four recordings captured signals made within colonies in response to *V. soror* workers (13–23 September; two 60-min recordings of attacks and two 5–6-min recordings immediately after hornet departure). Nine recordings captured colony response to *V. velutina* workers (7 September–9 October; nine 60-min recordings of attacks). Six recordings captured the response of colonies to extracts from the VGs of *V. soror* workers (29 August–20 September; five 60-min recordings and one 30-min recording). For this treatment, a 1 cm$^2$ piece of filter paper that had been impregnated with VG extract was pinned in the middle of each hive entrance, according to previously described methods [82]. Briefly, extracts were created by immersing VGs from three *V. soror* workers overnight in a vial of ether, then a piece of filter paper was repeatedly dipped in the vial mixture and dried before presentation to a colony. Finally, two control treatments captured signalling in colonies that were not visited by hornets prior to recording that day. The first no-attack control was an ether sham, for which an ether-only filter paper was pinned in the entrance of a hive (8–15 September; two 60-min recordings and one 53-min recording). The second no-attack control was free of manipulation (25 August–20 September; ten 5–22-min recordings).

To make an audio recording in a hive, a flat-frequency lavalier microphone (frequency range 50–15 000 Hz; Mediamart Joint Stock Company, Hanoi, Vietnam; no model number) was placed between frames as centrally to a colony's cluster as possible in wooden hives or among the cluster of bees on the bottom frame in observation hives (adjacent to the entrance). Microphones were inserted into hives either the day prior or the morning of recording to allow colonies time to settle down before observation. In the case of natural hornet attacks, we put separate microphones into several hives that were identified by beekeepers as being harassed by hornets of either species, with the hope that we would capture future attacks. To record colony soundscapes, a microphone was plugged into a digital recorder (Sony ICD-UX533F; 44.1 kHz sampling frequency, 16-bit, 50–20 000 Hz frequency, mp3 format).

## 2.3. Analysing worker-produced signals from audio within hives

Worker-produced signals on all audio recordings were characterized using Raven Pro (v. 1.5 [102]). Recordings were converted from mp3 to wav format in Raven Pro, then spectrograms for each recording were generated using a short-time Fourier Transform method with the following settings: brightness = 65%; contrast = 75%; spectrogram window size = 1200 points (21.5 Hz resolution); smoothing = on; colour map = hot; 50% overlap; Hann window function. All spectrograms were independently examined by two observers (HRM and HGK) who manually identified selections (signal duration and frequency boundaries) for each worker-produced signal. A merged dataset was subsequently generated that included selections made by both observers (n = 29 640 signals). At this stage, each signal was tentatively categorized as either a hiss or a pipe; pipes were further categorized as stop signals, antipredator pipes or long pipes based on signal traits (see Results for defining criteria for each signal type). Because antipredator pipes have not been previously described, three observers independently re-examined spectrograms of all audio recordings in the blind and identified antipredator pipes based on the signal traits that guided their categorization in the merged dataset (HGK and two assistants; see Acknowledgements). For a signal to be categorized as an antipredator pipe in this final dataset, at least two of the three observers had to independently identify them as such.

Using this final dataset of categorized signals, we calculated the number of each signal type that was produced per minute in each colony replicate. These values were averaged across minutes to determine mean rates of signalling per minute for each colony (for each signal type separately and for all signal

types combined). Colony means were then used to compare signalling rates across hornet-attack scenarios.

This final dataset was also used to determine mean signal durations for hisses and the three types of pipes, which were compared across hornet-attack scenarios. To examine mean fundamental frequencies of types of pipes, we identified a subset of clear pipes in the final dataset for which fundamental frequency was not obscured by other signals or background noise. We did this by first identifying pipes characterized in Raven Pro as having a maximum frequency (i.e. fundamental frequency) between 350 and 700 Hz, which is the approximate range for *A. cerana* pipes [51], and subsequently confirming through manual inspection of spectrograms that the maximum frequency for a target pipe corresponded to its lowest harmonic. This process yielded a subset of 761 pipes that were used to compare mean fundamental frequencies for each type of pipe across hornet-attack scenarios ($n = 502$ stop signals; $n = 96$ antipredator pipes; $n = 163$ long pipes). To improve the resolution of maximum frequency estimates for this subset of pipes, we increased the spectrogram window to 2100 (resolution of 10.7 Hz) and used the resulting maximum frequencies in all calculations of mean fundamental frequency. Control colonies made too few stop signals and antipredator pipes to be included in comparisons across hornet-attack scenarios, so they were dropped from that analysis (excluded $n = 9$ signals).

Finally, we identified periods during naturally occurring *V. soror* and *V. velutina* attacks when hornet workers were or were not in front of hives (based on videos; see next section). For each signal type, we determined how mean signalling rate changed per colony depending on the presence or absence of hornet predators for the two hornet species.

## 2.4. Analysing worker activities in videos of hive fronts

Each time we recorded audio in a colony, we also recorded a video of its hive front to capture the defensive activities of its workers (control replicates in observation hives had side views recorded). Clocks for videos were synchronized with corresponding audio recordings using a verbal mark that was announced at the start of both recordings. An additional three 60-min-long videos of *V. soror* attacks were included in this analysis to increase the sample size for this attack type; these colonies were part of the study, but their audio recordings failed. The first two colonies were recorded in Apiary 1 on 22 August and the third colony was recorded in Apiary 3 on 5 October.

We positioned a high-definition digital video camera (Sony Handycam HDR-PJ340) in front of a hive so that it was close enough to count individual bees on the hive front, but far enough away that arrivals and departures of hornets were visible (1–2 m, depending on the colony). We determined the number of hornets present per minute in each recording (and confirmed that hornets were not present when controls were recorded). At the start of each minute, we also counted the total number of bees on the hive front and the number of bees moving their antennae and mandibles over the hive surface (the behaviour of workers either applying faecal spots or manipulating previously applied spots; [82]). Counts made during each minute of a recording were used to calculate mean counts per colony, which were compared among all hornet-attack scenarios. We also determined the number of faecal spots added to the hive fronts of colonies that were exposed to hornet-attack stimuli (real *V. soror* workers, their VG extracts, or real *V. velutina* workers). For this comparison, colonies were included in the analysis only if they were recorded for 60 min to allow time for a spotting response to be initiated. Finally, we determined the number of group displays of body shaking (greater than or equal to three workers shaking their bodies simultaneously then stopping) during every minute that a hornet was present in front of a hive. Counts per minute were used to calculate colony means, which were compared between *V. soror* and *V. velutina* attack scenarios.

## 2.5. Statistical analysis

ANOVAs and *t*-tests were conducted using SAS (SAS Institute, v. 9.3). One-way ANOVAs were used to compare per colony signalling rates, worker counts and changes in faecal spots across hornet-attack scenarios. To avoid pseudoreplication, one-way ANOVAs with subsampling were used to compare signal duration and fundamental frequency across hornet-attack scenarios because multiple signals per category were characterized per colony. Two-way ANOVAs were used to compare signalling rates depending on hornet presence or absence outside colonies and hornet species. Where treatment effects or their interactions were significant, means were separated using Tukey's HSD *post hoc* tests. Two-tailed *t*-tests compared mean duration of stop signals and antipredator pipes in colonies (paired

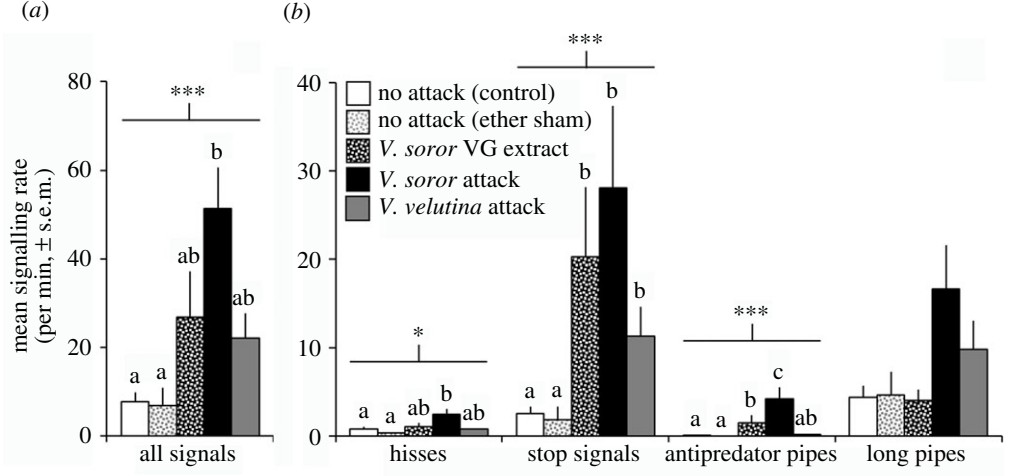

**Figure 1.** Attacks by *V. soror* generated the highest signalling rates in colonies. *A. cerana* colonies were recorded under different hornet-attack scenarios, including naturally occurring attacks by *V. soror* or *V. velutina* workers, exposure to VG extracts from *V. soror* workers (containing a marking pheromone that scouts use to target nests, in ether), and no-attack, hornet-free conditions (controls, including an ether sham). (*a*) Total signalling rate and (*b*) hiss and pipe signalling rates were compared across attack scenarios. Lowercase letters indicate significant differences between treatment means within each signal category; bars indicate comparison groups and asterisks indicate the significance of treatment effects (* $p < 0.05$; *** $p < 0.001$).

*t*-test), rate of antipredator signalling when single or multiple *V. soror* workers were present, and body shaking in response to *V. soror* and *V. velutina* attack (unpaired *t*-tests). All data were examined for normality and homogeneity of variances and, if assumptions were not met, log transformations were applied as needed. Equal variances among groups were typical. Normality was improved but not achieved in some transformed datasets. However, ANOVAs are expected to robustly control Type I error under the test conditions (i.e. large sample sizes and variance homogeneity [103]). The level of significance was set at $\alpha = 0.05$ for all tests.

## 3. Results

Colony soundscapes revealed a rich mix of signal types across the different attack scenarios that were examined for this study. We identified a total of 29 985 worker-produced signals in the 1307 min of recordings that were made inside *A. cerana* colonies. Of these, a small number of signals ($n = 345$ signals) were either too brief or obscured to be reliably characterized, so they were removed from the dataset. The remaining 29 640 signals were categorized as known or novel signal types. We describe how signal traits and rate of production changed in response to natural and simulated attacks by hornet predators, as well as simultaneous changes in defence-related worker activities around nest entrances.

### 3.1. Signalling rate increased dramatically in response to *V. soror* attacks

*Apis cerana* workers exchanged a cacophony of signals while they were under attack by hunting hornets and in the wake of recent visits by them. Attack by hornets strongly affected the total number of signals produced per minute in *A. cerana* colonies (one-way ANOVA: $F_{4,27} = 7.1$, $p = 0.001$). Mean signalling rate was highest in colonies that were attacked by *V. soror* workers and lowest in colonies that were not exposed to attack (figure 1*a*), representing a seven to eight-fold increase in signal production between these two extremes. Recordings of colonies experiencing active attacks by *V. soror* workers were noisy and frenetic, whereas recordings of control colonies were comparatively quiet and calm (electronic supplementary material, audio S1 versus audio S2). We likely underestimated signalling rate in colonies during *V. soror* attacks because at times colonies were so noisy that it was difficult to identify all signals in spectrograms. Mean signalling rates were intermediate for colonies that were attacked by *V. velutina* workers or exposed to extracts from the VG glands of *V. soror* workers, which are a source of a marking pheromone that scouts use to target prey nests (figure 1*a*). These recordings lacked the frenzy heard during *V. soror* attacks (electronic supplementary material, audios S3 and S4).

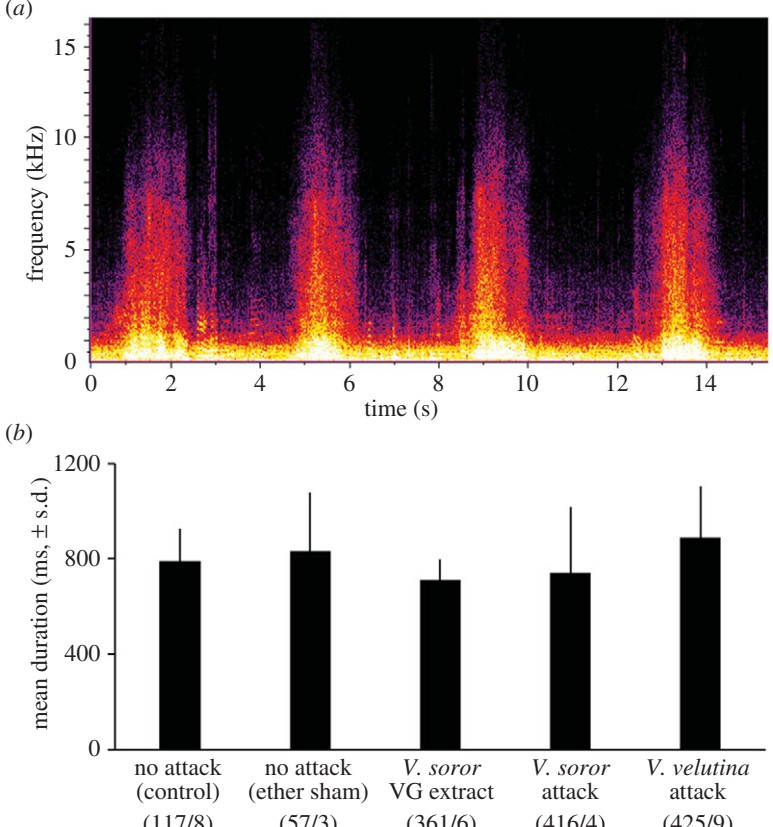

**Figure 2.** Hiss duration was similar across attack scenarios. Hisses were recorded in *A. cerana* colonies exposed to the hornet-attack scenarios described in figure 1. (*a*) Spectrogram of a series of four hisses, showing lack of harmonics and broadband energy across frequencies (listen to them with electronic supplementary material, audio S5). (*b*) Mean duration of hisses across attack scenarios. Parenthetical numbers indicate the number of hisses and colonies (i.e. subsamples/replicates) used to calculate treatment means.

## 3.2. Colonies produced diverse signal types in response to *V. soror* attack

We analysed changes in signal traits and signalling rates for two discrete types of worker-produced vibroacoustic signals—hisses and pipes—depending on the type of hornet-attack scenario that colonies experienced.

### 3.2.1. Hisses

Hisses were easily identified in recordings because they lacked discernible harmonics in spectrograms (examples: figure 2*a*; electronic supplementary material, audio S5 and video S1), as described previously for *A. cerana* [77]. We identified 1376 hisses in colony soundscapes, which comprised 4.6% of the total number of signals identified in the recordings. All colonies produced hisses at some point during the recording, except for two no-attack control colonies. The rate of hissing was highest on average in colonies that were attacked by *V. soror* workers, and significantly lower in control colonies; mean rate of hissing was intermediate for colonies attacked by *V. velutina* workers or exposed to VG extracts from *V. soror* workers (figure 1*b*; one-way ANOVA: $F_{4,27} = 3.3$, $p = 0.03$). Hiss duration was not influenced by the type of attack scenario to which colonies were exposed (figure 2*b*; one-way ANOVA with subsampling, attack type: $F_{4,35.2} = 1.4$, $p = 0.26$), although it differed among colonies that were recorded (subsample effect: $F_{25,1346} = 8.4$, $p < 0.0001$). Thus we determined average hiss duration for each colony and used those values to determine an overall mean of 802 ms ± 19 (s.d.) per hiss across the 30 colonies that produced them.

### 3.2.2. Pipes

Worker-produced pipes are universally described for *Apis* as having harmonic structure [31,48,51,55,56,64]. Thus, we categorized any recorded signal as a pipe if it showed harmonic structure

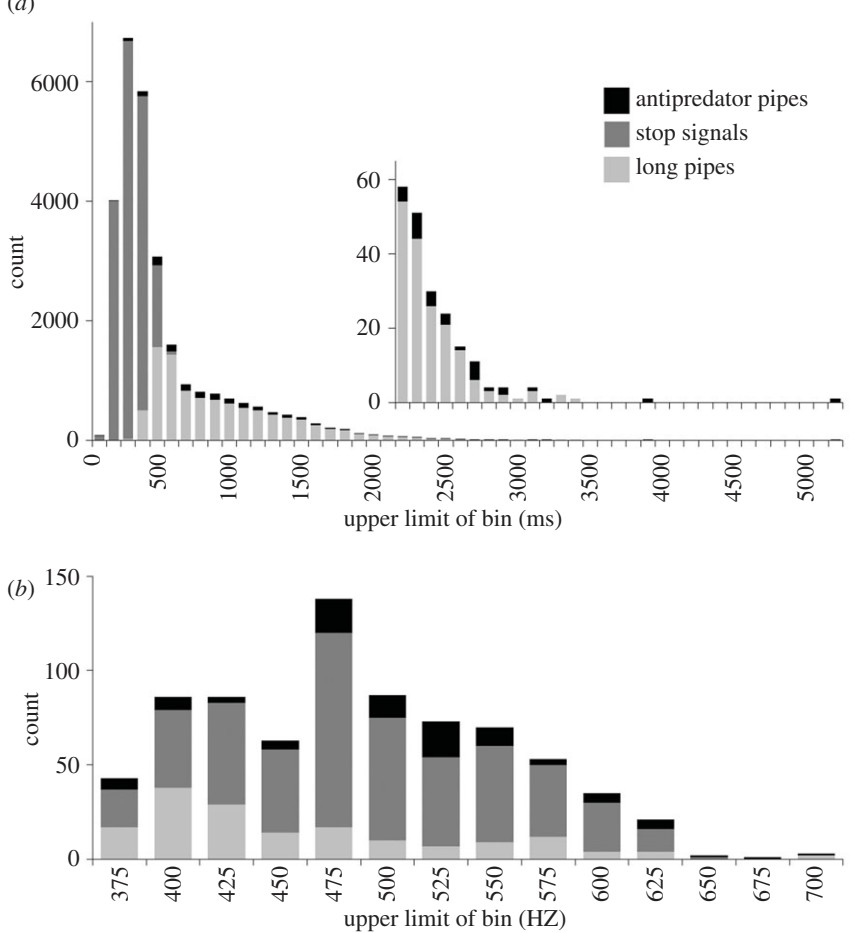

**Figure 3.** Duration and fundamental frequency varied among worker-produced pipes. Pipes were recorded in *A. cerana* colonies exposed to the hornet-attack scenarios described in figure 1. Pipes were identified by their harmonic structure in spectrograms. Histograms are provided for pipe (*a*) duration, *n* = 28 264 pipes distributed among 100 ms bins, and (*b*) fundamental frequency, *n* = 761 pipes distributed among 25 Hz bins; the upper limit of each bin is the value shown on the *x*-axes. Inset (in *a*) displays counts for bins directly below that are not visible due to the scale of the main *y*-axis.

in the corresponding spectrogram. Furthermore, we assumed that all pipes were produced by workers because we did not observe the characteristic pipes that *A. cerana* queens exchange during swarming [104,105] and there was no swarming in our apiaries, which mostly occurs outside of our study period in northern Vietnam (i.e. March–July [106]). In total, we identified 28 264 piping signals in all colony recordings, which constituted 95.4% of the signals in our dataset. However, there was considerable variability in the nature of pipes that workers produced, including metrics such as duration, fundamental frequency and frequency modulation (change in frequency over the duration of a signal [64]). Duration of pipes ranged from 56 ms at the shortest to 5145 ms at the longest, although 99% of pipes were less than or equal to 2 s (figure 3*a*). Fundamental frequencies of the majority of pipes were not discernible because they were either undetectable against background noise in colonies or difficult to separate from co-occurring signals. However, we were able to determine fundamental frequencies for 761 pipes, 90% of which fell between 400 and 600 Hz (figure 3*b*). From the total pool of pipe signals, we identified stop signals and antipredator pipes based on signal traits, with remaining pipes broadly categorized as 'long pipes'.

### 3.2.3. Stop signals

We identified stop signals based on traits that have been reported for them previously. Stop signals are described as notably brief in *Apis* with minimal to smooth frequency modulation and clear harmonic structure [40,51,54,56–58,64]. Head-butts associated with signal delivery help to identify stop signals

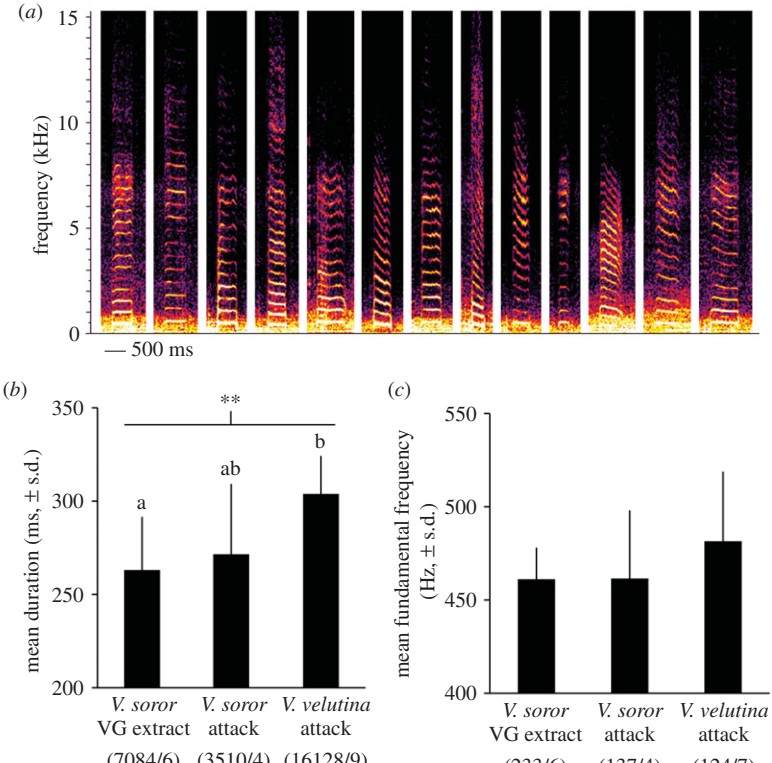

**Figure 4.** Stop signals were shorter in response to *V. soror* attack stimuli. Stop signals were recorded in *A. cerana* colonies exposed to the hornet-attack scenarios described in figure 1; sample size limited analysis to colonies exposed to real hornet attacks or VG extracts from *V. soror* workers. (*a*) Exemplars of stop signals taken from spectrograms (listen to them with electronic supplementary material, audio S6). (*b*) Mean duration of all stop signals produced in colonies exposed to hornet-attack stimuli. (*c*) Mean fundamental frequency of stop signals produced in colonies exposed to hornet-attack stimuli based on the subset of signals for which this signal trait was detectable in spectrograms. Lowercase letters indicate differences between treatment means where significant effects were found; the bar indicates the comparison group and asterisks indicate the significance of treatment effect (** $p < 0.01$). Parenthetical numbers indicate the number of stop signals and colonies (i.e. subsamples/replicates) used to calculate treatment means.

[55,64], but most workers (greater than 90%) also transmit stop signals by pressing their thorax to the comb or onto other workers [56]. Because we could not observe stop-signal delivery, we relied on their characteristic acoustic traits to identify them, as did Tan *et al.* [51]. We categorized pipes as stop signals if their duration was less than or equal to 500 ms and they lacked rapid or substantial frequency modulation, based on Tan *et al.* [51] who reported that stop signals in *A. cerana* colonies had a range in duration of 49–502 ms per signal and relatively even frequency. We categorized pipes greater than 500 ms as stop signals if they were clearly part of a stop-signal series, but this accounted for only 0.3% of all stop signals. Conversely, 12.1% of pipes less than or equal to 500 ms were not categorized as stop signals (figure 3*a*), primarily because they showed frequency modulation that differed markedly from published exemplars [51,56,64]; see §3.2.4 and §3.2.5 for how these pipes were categorized.

Using these criteria, we identified 17 411 stop signals in our recordings, which constituted the majority (62%) of the pipes in our dataset (figure 4*a*; electronic supplementary material, audio S6). Although stop signals were produced in every colony, they were observed at low rates in the absence of hornet attack (figure 1*b*). By contrast, rate of stop signalling was higher when colonies were exposed to hornet-attack stimuli (figure 1*b*; one-way ANOVA: $F_{4,27} = 11.5$, $p < 0.0001$). Because stop signals were produced relatively infrequently in control colonies, we analysed their traits in colonies that experienced hornet-attack stimuli only. Duration of stop signals differed significantly depending on type of attack and colony (figure 4*b*; one-way ANOVA with subsampling; attack type: $F_{2,16.4} = 8.7$, $p = 0.003$; subsample effect: $F_{16,16703} = 63.3$, $p < 0.0001$). Stop signals were shorter in response to *V. soror* stimuli and longer in response to *V. velutina* (figure 4*b*). Fundamental frequency was not affected by attack type, but differed among colonies (figure 4*c*; one-way ANOVA with subsampling; attack type:

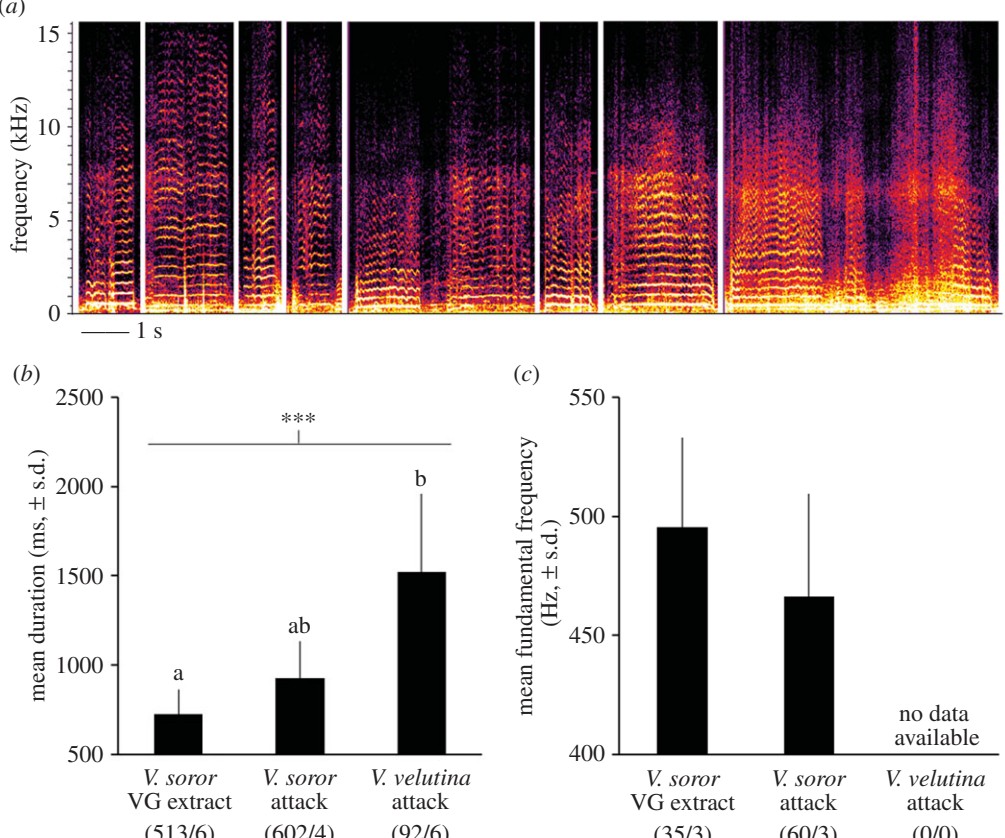

**Figure 5.** Antipredator pipes in response to hornet-attack stimuli. Antipredator pipes were recorded in *A. cerana* colonies exposed to the hornet-attack scenarios described in figure 1; limited sample size prevented the inclusion of the two hornet-free, no-attack controls in the analyses. (*a*) Exemplars of antipredator pipes taken from spectrograms (see electronic supplementary material, figures S1–S3 and audios S7–S9 for additional spectrograms and associated audio). (*b*) Mean duration of all antipredator pipes produced in colonies exposed to hornet-attack stimuli. (*c*) Mean fundamental frequency of antipredator pipes produced in colonies exposed to *V. soror* attack stimuli based on the subset of signals for which this signal trait was detectable in spectrograms. Lowercase letters indicate differences between treatment means where significant effects were found; the bar indicates the comparison group and asterisks indicate the significance of treatment effect (*** $p < 0.001$). Parenthetical numbers indicate the number of antipredator pipes and colonies (i.e. subsamples/replicates) used to calculate treatment means.

$F_{2,29.8} = 0.6$, $p = 0.57$; subsample effect: $F_{14,477} = 3.9$, $p < 0.0001$). Thus, we calculated an overall mean fundamental frequency of $469 \pm 31$ (s.d.) Hz for stop signals using colony averages.

### 3.2.4. Antipredator pipes

A subset of pipes differed from stop signals because of strong, irregular and strikingly rapid modulation of frequency over the duration of each signal, as evidenced in spectrograms (figure 5*a*, electronic supplementary material, figures S1–S3 and audios S7–S9). They showed harmonic structure coupled with broadband energy under 10–12 kHz. These pipes were produced at the highest rates in colonies that were attacked by *V. soror* workers, at lower rates in colonies that were attacked by *V. velutina* workers or exposed to the VG extracts from *V. soror* workers, and rarely in control colonies (figure 1*b*; one-way ANOVA: $F_{4,27} = 15.1$, $p < 0.0001$). We labelled these signals as 'antipredator pipes' because they were produced specifically in the context of exposure to predator stimuli (either real hornets or their VG extracts).

We videorecorded multiple workers making antipredator pipes. Several workers produced antipredator pipes on the front of a hive that had paper impregnated with VG extract pinned in its entrance (electronic supplementary material, videos S2–S4). Workers that produced antipredator pipes acted similarly: each worker raised her abdomen dorsally and ran rapidly while buzzing her wings,

exposing her Nasonov gland repeatedly as she moved. The paths of piping workers at times were seemingly haphazard or zig-zagging around the entrance; at other moments workers quickly approached and then retreated from the pinned paper while piping. Other antipredator pipes were recorded concurrently inside the hive. In a preliminary trial, we placed four *V. soror* workers in a screened enclosure attached to the front of a hive. In a video taken 5 min after the enclosure was removed, we recorded a worker making antipredator pipes and exposing her Nasonov gland as she moved from inside the hive to the landing board outside the entrance, while other non-piping workers exposed their Nasonov glands or applied faecal spots around the nest entrance (electronic supplementary material, video S5). Finally, we observed workers attempt to bee ball a *V. soror* worker immediately after she killed a nestmate that was making antipredator pipes at the entrance (electronic supplementary material, video S6).

In total, we identified 1213 antipredator pipes in colonies, constituting 4% of pipes in our dataset. A quarter (24.6%) of antipredator pipes were under 500 ms, but the majority were substantially longer (figures 3a and 5a, electronic supplementary material, figures S1–S3; mean duration $886 \pm 514$ ms (s.d.) and range 98–5145 ms per pipe, measured from the start to end of visible harmonics). Accordingly, longer mean signal duration differentiated antipredator pipes from stop signals when values were compared across colonies in which both signal types were recorded (paired *t*-test: $t_{15} = 14.2$, $p < 0.0001$). Antipredator pipes were shorter in colonies exposed to *V. soror* attack stimuli compared to *V. velutina* attacks (figure 5b; one-way ANOVA with subsampling; attack type: $F_{2,35.2} = 12.8$, $p < 0.0001$; subsample effect: $F_{13,1191} = 3.1$, $p = 0.0001$). Because of the timing of when these pipes were produced (see below), the fundamental frequency was detectable for only a small subset of antipredator pipes made in colonies exposed either to real *V. soror* workers or their VG extracts ($n = 95$ signals); the fundamental frequency of antipredator pipes did not differ between these scenarios (figure 5c; one-way ANOVA with subsampling; attack type: $F_{1,3.8} = 1.4$, $p = 0.31$; subsample effect: $F_{3,90} = 1.9$, $p = 0.14$). We determined a mean fundamental frequency of $481 \pm 40$ Hz (s.d.) for antipredator pipes by averaging colony means across both treatment groups.

### 3.2.5. Long pipes

After categorizing stop signals and antipredator pipes, 9640 pipes remained in our dataset (34% of all pipes). These pipes were generally long, although shorter signals were part of many long-pipe series (range 196–3339 ms per pipe). They were recorded in all colonies except one no-attack control colony. In general, these long pipes differed from antipredator pipes in that they lacked the latter's rapid and unpredictable modulation in frequency; when frequency changed over the duration of these pipes, it did so in smooth sweeps (figure 6a; electronic supplementary material, audio S10). Their acoustic structure in spectrograms did not suggest that subcategories of long pipes existed and trends across different attack scenarios were weak. The signalling rate of long pipes was not affected by attack type (figure 1b; one-way ANOVA: $F_{4,27} = 2.6$, $p = 0.06$) and neither was signal duration (figure 6b; one-way ANOVA with subsampling; attack type: $F_{4,27.4} = 1.7$, $p = 0.17$; subsample effect: $F_{26,9609} = 88.2$, $p < 0.0001$). There was a marginally significant effect of attack type on fundamental frequency, although differences were not strong enough to separate means (figure 6c; one-way ANOVA with subsampling; attack type: $F_{4,20.7} = 3.0$, $p = 0.043$; subsample effect: $F_{16,142} = 3.0$, $p = 0.0003$).

## 3.3. Colonies produced antipredator pipes when *V. soror* workers were present at nests

We compared rates of hissing and piping during natural attacks that included periods when hornets were either in front of hive entrances or had left the vicinity of the hives. This dataset included 60-min recordings of two *V. soror* attacks and nine *V. velutina* attacks (hornets were present in front of hives for 5–14 min and 9–52 min, respectively). Only antipredator pipes were affected by the presence of hornets; they were produced at high rates when *V. soror* workers were outside entrances, but otherwise their production was relatively low, including when *V. velutina* attackers were present (figure 7; two-way ANOVA; hornet presence: $F_{1,18} = 53.9$, $p < 0.0001$; hornet species: $F_{1,18} = 235.9$, $p < 0.0001$; interaction: $F_{1,18} = 47.5$, $p < 0.0001$). Rates of antipredator piping were similar whether there was a lone *V. soror* worker or multiple workers attacking (mean $18 \pm 5$ versus $20 \pm 3$ antipredator pipes/min, respectively; *t*-test: $t_{15} = 0.3$, $p = 0.74$). By contrast, the presence of either hornet species in front of hives did not affect signalling rates for stop signals (figure 7; two-way ANOVA; hornet presence: $F_{1,18} = 0.4$, $p = 0.56$; hornet species: $F_{1,18} = 1.8$, $p = 0.20$; interaction: $F_{1,18} = 0.7$, $p = 0.43$) or long pipes (figure 7; two-way ANOVA; hornet presence: $F_{1,18} = 0.2$, $p = 0.70$; hornet species: $F_{1,18} = 0.1$, $p =$

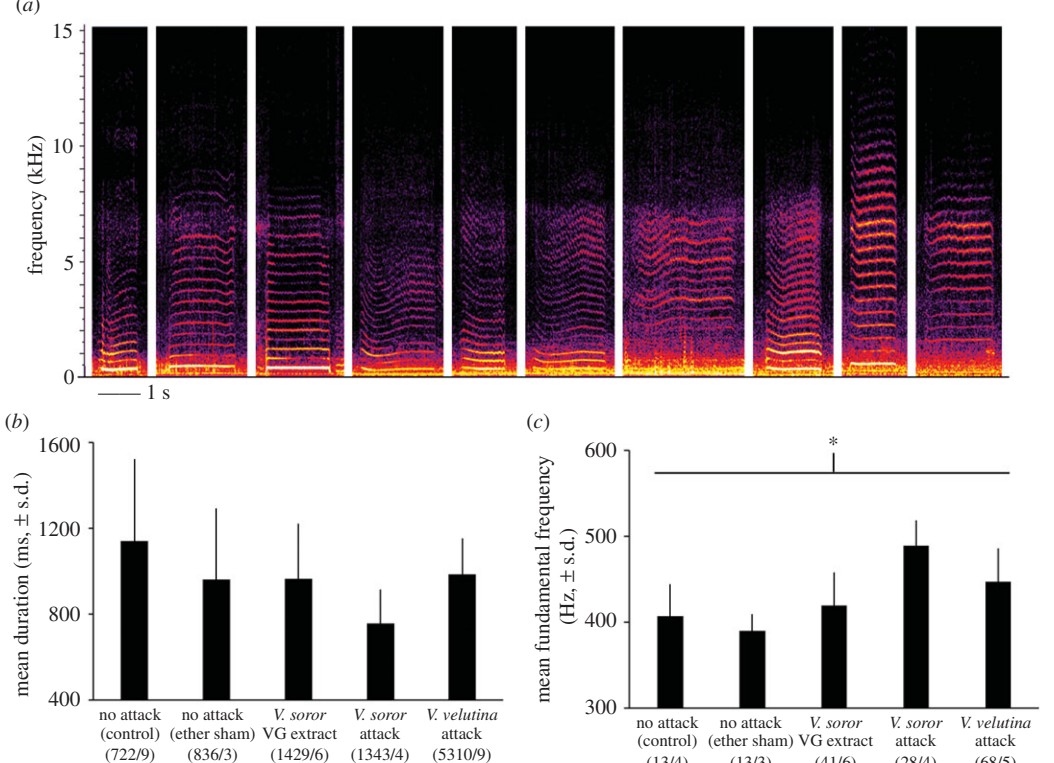

**Figure 6.** Most colonies produced long pipes with variable signal traits. Long pipes were recorded in *A. cerana* colonies exposed to the hornet-attack scenarios described in figure 1. (*a*) Exemplars of long pipes, i.e. pipes that were not categorized as stop signals or antipredator pipes, taken from spectrograms (listen to them with electronic supplementary material, audio S10). (*b*) Mean duration of all long pipes recorded in five attack scenarios. (*c*) Mean fundamental frequency of long pipes in the same attack scenarios, but based on the subset of signals for which this signal trait was detectable in spectrograms. The bar indicates the significant comparison group and the asterisk indicates the significance of the treatment effect (* *p* < 0.05); a weak treatment effect resulted in an inability to separate means with a Tukey's HSD *post hoc* test. Parenthetical numbers indicate the number of antipredator pipes and colonies (i.e. subsamples/replicates) used to calculate treatment means.

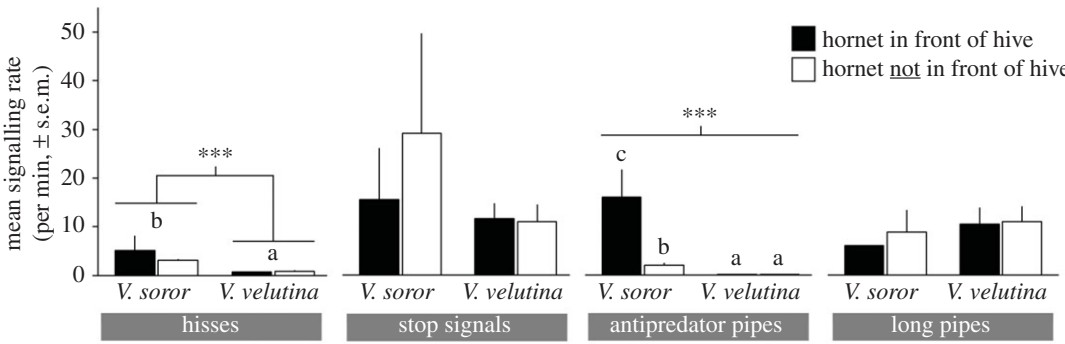

**Figure 7.** Colonies produced antipredator pipes when *V. soror* attackers were outside hives. Worker-produced signals were recorded in *A. cerana* colonies that were attacked by either *V. soror* or *V. velutina* workers. Replicates included 60 min audio recordings that had intervals with and without hornets present outside the front of hives (*n* = 2 *V. soror* attacks; *n* = 9 *V. velutina* attacks). For each signal type (bottom), mean signalling rate was compared between hornet species and hornet presence, as determined from simultaneously recorded videos of hive fronts (including entrances). Bars indicate comparison groups; lowercase letters indicate significant differences between means for each signal type; asterisks indicate significance level of comparison (*** *p* < 0.0001).

0.79; interaction: $F_{1,18} = 0.01$, $p = 0.91$). Finally, hissing rate was not affected by the presence of hornet attackers in front of hives, although colonies generally hissed more when exposed to *V. soror* workers compared to *V. velutina* workers (figure 7; two-way ANOVA; hornet presence: $F_{1,18} = 0.4$, $p = 0.55$;

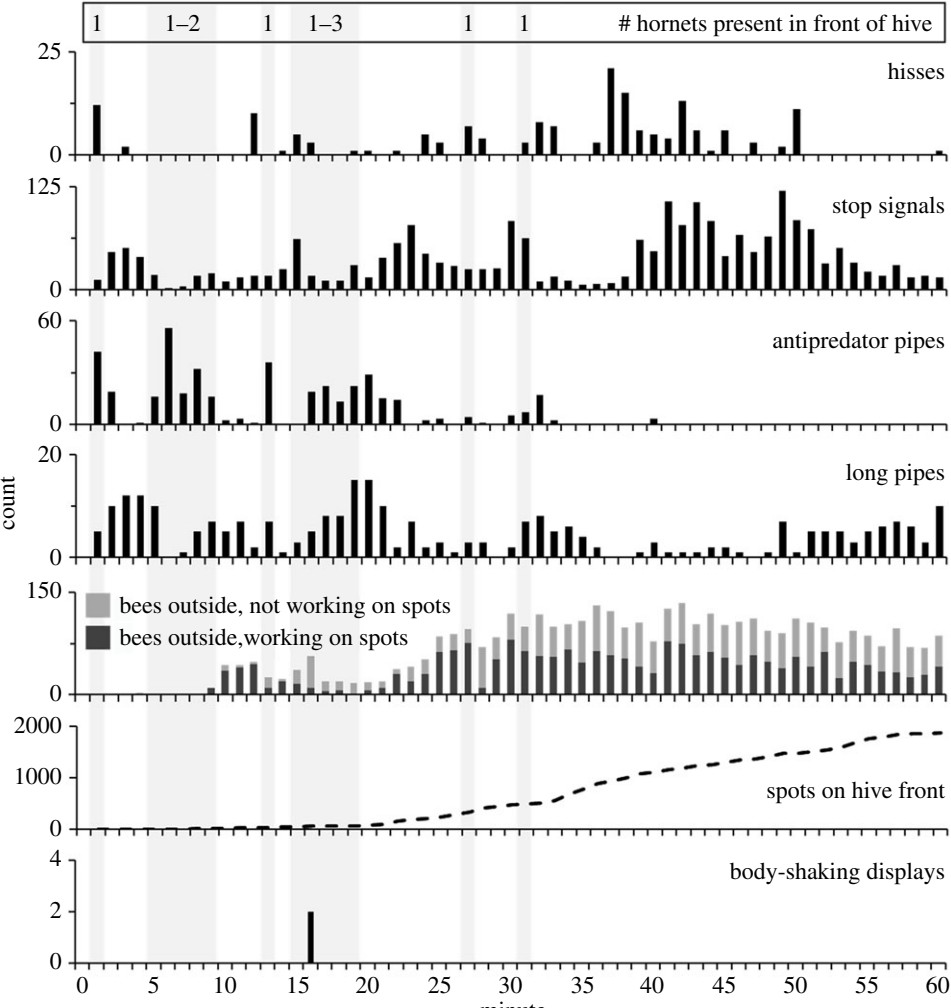

**Figure 8.** Timeline of signalling and defence responses of *A. cerana* workers during a multiple-hornet *V. soror* attack. A *V. soror* attack on an *A. cerana* colony was recorded for 60 min (audio inside colony and video of hive front), during which time 1–3 *V. soror* workers (indicated at top) repeatedly flew in front of the hive and approached its entrance (attack intervals are indicated by grey vertical bars). Total counts per minute of worker-produced hisses and pipes are shown, as are counts per minute of the number of bees on the hive front (working or not working on spots), the number of faecal spots on the hive front and the number of body-shaking displays by workers.

hornet species: $F_{1,18} = 26.7$, $p < 0.0001$; interaction: $F_{1,18} = 0.5$, $p = 0.51$). Minute-by-minute examples of signal production in colonies showed how the occurrence of antipredator pipes aligned with the presence of *V. soror* workers, but not *V. velutina* workers, as natural attacks unfolded (figure 8, electronic supplementary material, figures S4 and S5). Antipredator pipes were also produced consistently over time when a colony was exposed to a marking pheromone (VG extracts) of *V. soror* at its hive entrance (electronic supplementary material, figure S6).

## 3.4. Colonies employed different defences depending on type of hornet attack

We monitored the activities of *A. cerana* workers around colony entrances to determine whether specific defensive behaviours co-occurred with signalling within hives. Attacks by hornet workers of either species yielded the highest number of bees on hive fronts per minute, based on averaged counts made at the start of each minute of recording for each colony (table 1; one-way ANOVA: $F_{4,24} = 8.8$, $p = 0.0002$). If attacked by *V. soror* workers, about half of these workers had their heads and antennae oriented downward, presumably manipulating faecal spots on their hive's surface; this behaviour was observed infrequently for colonies in other hornet-attack or control scenarios

**Table 1.** *Apis cerana* colonies responded to *V. soror* attacks with faecal spotting and *V. velutina* attacks with body shaking. The defence activities of workers outside nest entrances were determined from video recordings of hive fronts taken while audio recordings were made inside hives. Colonies were recorded under different hornet-attack scenarios, including naturally occurring attacks by *V. soror* or *V. velutina* workers, exposure to VG extracts from *V. soror* workers (containing a marking pheromone that scouts use to target nests, in ether), and no-attack, hornet-free conditions (controls, including an ether sham). The number of workers on the hive front and the number of workers working on hive surfaces (head and antennae oriented downward) were determined for all colonies at the start of each minute in each recording. Change in spot number was determined only for colonies that were exposed to hornet-attack stimuli and recorded for 60 min to allow time for a faecal-spotting response. Body shaking per minute was assessed only for treatments that involved attacks by real hornets, and only during minutes when hornets were present in front of hives. Lowercase letters indicate significant differences among treatment means.

| activity on hive front (mean ± s.e.m.) | no attack (control) | no attack (ether sham) | *V. soror* VG extract | *V. soror* attack | *V. velutina* attack |
|---|---|---|---|---|---|
| number of workers on hive front (per min) | $4 \pm 2^a$ | $6 \pm 2^{a,b}$ | $1 \pm 0.4^a$ | $32 \pm 9^b$ | $20 \pm 3^b$ |
| number of workers working on surface (per min) | $0 \pm 0^a$ | $0 \pm 0^a$ | $0.1 \pm 0.1^a$ | $11 \pm 5^b$ | $0 \pm 0^a$ |
| number of spots added (change over 60 min of observation) | n.a. | n.a. | $10 \pm 6^a$ | $600 \pm 321^b$ | $1 \pm 1^a$ |
| number of group displays of body shaking (per min) | n.a. | n.a. | n.a. | $3 \pm 2^a$ | $22 \pm 3^b$ |

(table 1; one-way ANOVA: $F_{4,24} = 13.7$, $p < 0.0001$). Accordingly, when colonies that were recorded for 60 min were considered, only colonies attacked by *V. soror* workers had a substantial number of spots applied to hive fronts by the end of the hour (table 1; one-way ANOVA: $F_{2,15} = 50.5$, $p < 0.0001$). *Vespa velutina* attacks drew *A. cerana* workers outside, but they did not attend to spots to hive fronts. Rather, they responded with more frequent body shaking when *V. velutina* workers were in front of hives, a behaviour that was rarely observed when *V. soror* workers were present (table 1; t-test: $t_9 = 4.6$, $p = 0.0006$).

## 4. Discussion

This study deeply explores the diversity of vibroacoustic signals that are produced in *A. cerana* colonies and the pronounced shifts in signalling that occur when colonies are attacked by hornets. These signals consisted of hisses, generated by many workers collectively, and several types of pipes produced by individual workers. Our dataset of almost 30 000 signals revealed that a colony is a consistently bustling signalling space, even when it is seemingly unperturbed. However, during periods of extreme duress, such as threat from predatory hornets, rates of signalling ratchet up considerably. Most notably, signalling responses were the strongest when colonies were assailed by *V. soror*, the largest and deadliest hornet predator in Vietnam. When faced with this predator, the primary ways that worker signalling intensified were by a dramatic increase in the rate of production of hisses and stop signals, as well as the initiation of 'antipredator pipe' signalling, a previously undescribed type of worker pipe that was produced almost exclusively when colonies were exposed to attack stimuli from *V. soror* (i.e. hornets or their gland extract). Naturally occurring attacks by *V. soror* workers resulted in a seven-fold increase in overall signalling rate compared to no-attacks conditions. By contrast, attacks by *V. velutina* workers generated about half the rate of signalling that *V. soror* attacks elicited. It is also clear from these colony soundscapes that hisses, stop signals and longer (non-antipredator) pipes were routinely produced by colonies outside of periods of obvious stress, albeit at lower levels. Workers are constantly communicating with each other, in both good times and in bad, but antipredator signal exchange is particularly important during dire moments when rallying workers for colony defence is imperative.

One of the most perilous moments an *A. cerana* colony faces is the arrival of a giant hornet predator at its nest. The size and strength of giant hornets equip them for launching devastating group attacks on other social insect colonies. Remarkably for an insect, they are at the apex of their food web [83]. Thus, it is not surprising that *A. cerana* has evolved signals to cope with the menace that giant hornets pose. Notably, the production of antipredator pipes peaked during the acutely hazardous periods when *V. soror* workers were visible directly outside of colony entrances. The term 'antipredator pipe' suggests that they are shaped by natural selection as part of the suite of alarm signals used by *A. cerana* colonies when nests are threatened by predators. However, in the absence of more information about the function of antipredator pipes (see below), this label does not describe sender intent, receiver response, or a link to a particular predator or means of defence [6]. This acoustic signal likely evolved traits that get the attention of recipients, similar to the abrupt-onset pulses, noisy broadband frequency spectra and dramatic frequency modulations that characterize the alarm shrieks, fear screams and panic calls of primates, birds and meerkats [6,13,14,107–109]. Moreover, workers appear to string together individual antipredator pipes (which is what we measured) into longer, insistent messages, analogous to how individual clangs of a bell contribute to a sustained fire alarm.

Audio recordings support the impression that antipredator pipes are an alarm signal. Even more convincing are videos of workers producing antipredator pipes on a hive front after exposure to VG extracts from *V. soror* workers. These workers raced around frantically, approaching and retreating from the extract-soaked paper, piping repeatedly and in quick succession. When signalling, a worker raised her abdomen, exposed her Nasonov gland, and buzzed her wings while standing in place or running. The vibration of the wings and thorax is a key element of other well-studied worker and queen pipes, coupled with the sender pressing her thorax into a substrate to transmit the signal to nestmates [29,31,47,55,56]. When antipredator pipes were audible in videos (electronic supplementary material, videos S2–S5), workers creating them held their wings far apart and beat them with visible amplitude, which differs from the almost imperceptible movement (to the human eye) and closely held position of wings for pipes made in other contexts [29,31,47]. The lunging and darting run and clear wingbeats of antipredator pipers is akin to the scrambling, buzzy movement of buzz-runners in swarms, although buzz-runs lack the clear harmonics we saw in antipredator pipes [110]. Thus, antipredator pipes share production elements in common with pipes that have harmonic structure, although wing buzzing by antipredator pipers might contribute to the broadband energy of their signals. It was not clear from our videos whether workers pressed their thoraces onto hive surfaces as they piped. Furthermore, we do not know how antipredator pipes were transmitted to recipients within hives, thus delivery and mode of perception remain open questions.

The function of antipredator pipes requires confirmation. We have described their vibroacoustic characteristics and how they are made by workers, but to understand the evolution of antipredator pipes as a signal, we need to know the information they convey to recipients and their response to them [1]. Based on the timing of their production, it is reasonable to speculate that, at a minimum, antipredator pipes inform nestmates about the presence of a hornet outside the nest. Intriguingly, our description of antipredator piping matches a 50-year-old description from Japan of *A. cerana* workers raising their gasters and vibrating their wings while approaching a *V. mandarinia* attacker prior to bee balling [70,83]. The signal traits of antipredator pipes also closely match those made by *A. mellifera cypria* when they try to ball *Vespa orientalis* [111], and may be the pipes heard when *A. mellifera liguistica* guards grapple with *Vespa simillima* at their nest entrances [47]. The sight and odours of real *V. soror* attackers are likely important for triggering strong antipredator piping and a movement of workers to hive fronts because VG extracts alone resulted in relatively weaker responses. For *A. cerana* in Vietnam, defences that require assembly, teamwork, and a degree of synchronization include bee balling [89,94], body shaking [99,100] and faecal spotting [82].

Antipredator pipes seem to be a rallying call for collective defence, but our study does not make clear whether they signify preparation for a specific type of defence. We observed balling of several hornet species at our field site (e.g. electronic supplementary material, video S6), but not during our recordings. Nonetheless, antipredator pipes might assemble workers for bee balling, the rapid initiation of which appears to be triggered by the presence of a live hornet and both bee and hornet alarm pheromones [51,112]. Electronic supplementary material, video S6, shows this sequence of events. It is unlikely that antipredator pipes convey a need to assemble a body-shaking defensive force at the entrance because colonies more often employed body shaking in response to hovering *V. velutina* attackers, despite low rates of antipredator piping. Conversely, we did not observe much body shaking in response to *V. soror* attacks, even though rates of antipredator piping were high. Finally, the production of antipredator pipes and the initiation of faecal spotting co-occurred in colonies that were

attacked by *V. soror* (figure 8, electronic supplementary material, figure S4 and video S5). However, follow-up studies are necessary to explore whether antipredator pipes, in addition to emergency waggle dances, are among the signals that prompt *A. cerana* workers to begin foraging for dung and other materials that repel hornets from entrances [82,113]. It is also possible that antipredator pipes rally workers to prepare for more than one of these defences. Playback experiments are necessary to resolve the response of workers to this signal.

Workers persistently exposed their Nasonov glands while making antipredator pipes, suggesting gland secretions are components of a multimodal defence signal. Nasonov volatiles are well known for their attraction and orientation function in *A. mellifera* [114,115]. While not as well studied in Asian honeybees, the Nasonov volatiles of *A. cerana* promote worker clustering in simple assays [116,117] and are involved in coordinating defences in other species [118,119]. Nasonov volatiles from antipredator pipers may prompt receivers in *A. cerana* colonies to amass near nest entrances as a defensive reserve. Such entrance assemblies are often reported across cavity-dwelling species of honeybees when workers collectively defend their colonies against sympatric hornet predators [47,70,72,88,90–93,118]. Thus, antipredator piping may be part of a multimodal signal that pairs vibrations, which have the advantage of speedily transmitting a pressing message ('predator attacking') [1,29], with a nonredundant chemical signal that orients defenders ('gather here') [20]. *Apis cerana*'s response parallels how the recruitment calls of meerkats or the mobbing calls of many birds assemble conspecifics to confront predators [120,121]. Like other mobbing calls, the signaller should be easy to locate based on signal traits [122]. Alternatively, antipredator pipes may be an 'on-guard' call that informs receivers of continuing danger [122]. Tactile cues from darting, agitated workers may activate quiescent nestmates for defence, like buzz-runners that rouse sluggish workers to lift off from swarm bivouacs [110,123]. Simultaneous use of vibroacoustic and chemical (and possibly tactile) signals could enhance or modulate the response of signal recipients, or produce a behaviour entirely different than what each component generates in isolation [19,20,124]. In combination, such a multimodal signal could attract more attention, more effectively convey information in a chaotic environment, or contain more information for recipients [19–21,125,126], increasing the likelihood that *A. cerana* workers respond quickly or with an appropriate course of action against an imminent hornet threat. This is a key topic for future exploration, given how understudied signalling is in *A. cerana* relative to *A. mellifera* and the limited number of investigations of multimodal antipredator signals for animals generally [20,127,128].

Antipredator pipes add a discrete signal to *A. cerana*'s known alarm repertoire of hisses and stop signals [51,77]. Discrete signal features help to avoid ambiguity, aligning with predictions of greater communicative complexity when species are highly social and have sophisticated predator defences [9,129,130]. While hisses are considered aposematic warning signals in many animal taxa, including bees [32,48,80,127,131–135], the role hisses play in *A. cerana* defence is not clear [77,80,136]. Hisses were part of how colonies responded to hornet attacks in our study, but colonies also hissed under no-attack conditions. This latter finding agrees with previous studies showing that *A. cerana* colonies hiss without apparent provocation [80,136]. Because of their timing in this study, it is unlikely that hisses are used to deter high-threat hornet predators. Only antipredator pipes showed this potential. Instead, hisses are probably directed at nestmates. Colonies become extraordinarily noisy when *V. soror* attackers arrive (electronic supplementary material, audio S1) and hisses may help to coordinate a collective response in the aftermath. Workers become still after hissing (electronic supplementary material, video S1; [77]), which may quieten a chaotic signalling environment enough to allow colony members to attend to other alarm signals produced by distressed nestmates. Similar to antipredator pipes, the production of stop signals also spiked dramatically when colonies were exposed to hornet stimuli. Our findings echo the reaction of *A. cerana* colonies to attack by *V. mandarinia*, including modification of stop-signal traits when attacked by giant hornets [51,52]. Our studies differed from those in how signal traits were modified, which may reflect signalling nuances revealed during naturally occurring hornet attacks (this study only), geographical differences in predator–prey interactions, or regional alarm signal dialects, as occurs in other animal systems [137–141]. Lastly, we recorded thousands of long, non-antipredator pipes in all treatment scenarios. It is difficult to say how these pipes fit into the signalling repertoire of *A. cerana* because they have not been reported in the literature. Variability among long pipes suggests that there is much to explore about the information senders convey via pipes that do not fit stop signal or antipredator pipe profiles.

This study robustly describes how *A. cerana* workers use a diverse repertoire of vibroacoustic signals to respond to naturally unfolding attacks by hornet predators. In other animals, such as primates, squirrels, birds and mongooses, alarm calling can be complex, encoding predator or attack type (i.e. predator-specific or functionally referential signals), level of urgency (i.e. affect or risk-based signals)

or both, which may result in different defensive or escape responses by recipients [9,12–16,120,142–148]. Alarm communication in *A. cerana* colonies should be considered within the same framework because of the complexity of their repertoire and the variety of predators that attack colonies [68], each of which may require the coordination of one or more defence strategies at the group level [8,149].

Predator-specific alarm signals are predicted to evolve when species have predator-specific defence strategies that can be employed by signal receivers [8,150]. Do *A. cerana* alarm signals convey predator-specific information to receivers? As a starting point, *A. cerana* colonies produce distinct categories of acoustic signals (hisses, antipredator pipes and stop signals), as do other animals with predator-specific alarm calls [12,13,151]. Traditionally, to have predator-specific meaning, an alarm signal should have informative value and referential specificity [8,150]. In this study, antipredator pipes best meet these criteria. They were reliably produced when *V. soror* workers were present, and rarely produced otherwise. By contrast, stop signals and hisses were produced in all treatment scenarios and at similar levels in attacked colonies whether hornets were present or absent, and both signals are used in other, non-predator contexts [56–58,63,80]. More recently, there is emphasis on contextual cues for refining signal meaning for receivers [145,152,153], so these signals may convey more predator-specific information when they are used during hornet attacks. They may also convey more specific information when combined, similar to how predator-specific signals are achieved in social mammals when non-referential signals are used together [15,144,154–157].

Do *A. cerana*'s alarm signals convey level of urgency? Rather than conveying referential information about the type of attacker that is encountered, signalling could reflect the level of fear or danger perceived by signal producers. In many mammal and bird species, rate of signalling can indicate the signaller's perception of predatory risk, which is often tied to predator proximity [13,16,142–145,147,156,158]. In this study, increased production rates of hisses, stop signals and antipredator pipes during *V. soror* attack could convey to colony members a high level of predatory risk that reflects the tendency of *V. soror* workers to closely approach nest entrances and attack in groups. In fact, antipredator pipes may be produced primarily in response to *V. soror* attackers simply because they frequently approach entrances and doggedly linger there as they hunt (this study; [70,82]), whereas *V. velutina* hawk bees at a distance (this study; see also [72,82,92]). Signalling rate is a particularly reliable indicator of urgency when other signal attributes are difficult to discern in a noisy environment [143], such as within *A. cerana* nests when *V. soror* workers attacked (electronic supplementary material, audio S1). However, our study does not untangle the attributes of *V. soror* attacks that drive changes in signal production. Further studies are necessary to discern whether colonies recognize attackers as giant hornets specifically or as large predators generally (both are predator-specific categories) or whether signal use is influenced by the tendency of *V. soror* attackers to hunt in groups and more closely approach nest entrances compared to smaller hornets (perceived risk or urgency).

*Apis cerana* colonies are a fascinating example of a eusocial insect whose rich use of alarm signals mirrors the features of sophisticated antipredator systems of socially complex vertebrates. Asian honeybees repel hornet predators with collective force and multiple antipredator strategies [11], highlighting the need for unambiguous alarm signals, shared among group members, to organize these efforts. In *A. cerana*, the dynamics of signal use, including the attention-grabbing traits of antipredator pipes, are multifaceted when each signal type is considered separately. The signalling landscape becomes substantially more complex if all of these signals are considered in unison. When a dangerous predator such as *V. soror* approaches a colony, workers produce several types of vibroacoustic signals at a frenetic pace and in parallel. These signals likely comingle with Nasonov and venom gland volatiles produced by alarmed nestmates, as well as hornet-produced alarm signals that eavesdropping honeybees use to influence their counterattack [51,89,112,159–161]. Visual cues from live, hunting hornets add to a cocktail of multimodal signals and cues to which workers must respond appropriately, and quickly, if their colony is to survive an attack by giant hornets.

Ethics. The field studies described in this paper involved observation of free-living animals. No protected or endangered species were sampled and we did not harm bees or hornets while observing their natural interactions. Gland extracts were obtained from hornets purchased from a commercial wasp farmer. We received permission to conduct our research from the beekeepers on whose property the study apiaries were located, as well as the People's Committee of Tây Đằng Town, Ba Vì District. Because the beekeepers had land-use rights on these properties according to Vietnamese law, no permits were required once permissions were granted.

Data accessibility. All data and SAS code can be found with electronic supplementary material.

The data are provided in the electronic supplementary material [162].

Authors' contributions. H.R.M. conceived and coordinated the study, acquired funding, conducted the investigation (fieldwork and analyses of recordings) and statistical analyses, and wrote and revised the manuscript; H.G.K.

conducted the investigation (analyses of recordings) and wrote and revised the manuscript; G.W.O. conceived and coordinated the study, acquired funding, conducted the investigation (fieldwork) and critically revised the manuscript; L.T.P.N. and H.D.P. acquired funding, conducted the investigation (fieldwork) and reviewed the manuscript; O.M.K. conducted the investigation (fieldwork) and reviewed the manuscript; N.T.P. conducted the investigation (fieldwork) and critically revised the manuscript.

Competing interests. The authors declare that they have no competing interests.

Funding. Funding for this study was provided by the National Geographic Society Committee for Research and Exploration (grant no. 9338-13; to G.W.O., H.R.M., L.T.P.N., H.D.P.; nationalgeographic.org), the Vietnam National Foundation for Science and Technology Development (grant no. 106.05-2018.303; to L.T.P.N.; nafosted.gov.vn) and Wellesley College (Knafel Chair in the Natural Sciences, the Science Center Summer Research Programme, the Sophomore Early Research Programme, the First-Year Apprentice Programme and the Library and Technology Services Open Access Fund; to H.R.M.). The funders had no role in study design, data collection and analysis, decision to publish or preparation of the manuscript.

Acknowledgements. We are indebted to Thanh Thong Nguyen, Dai Dac Nguyen and Duong Dinh Tran for capable field assistance in Vietnam. We thank Erica Maul, Patty Benitez-Lomi, McKenna Montminy, Sophia Peña, Zoe Glasser-Breeding and Cynthia Gomez for assisting over many hours with analyses of audio and video recordings at Wellesley College. Xuat Van Pham provided hornets from his vespiary for our studies. We are very grateful to Quan Viet Phung, Trung Van Phung, Dao Van Bui and Ngu Van Quach for hosting our research team in their apiaries. We appreciate feedback from the reviewers and editor during the review process; their input improved the clarity of the manuscript.

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
