## [Peer Review File · Royal Society Open Science]

Review History

RSOS-211215.R0 (Original submission)

Review form: Reviewer 1

Is the manuscript scientifically sound in its present form?

Yes

Are the interpretations and conclusions justified by the results?

Yes

Is the language acceptable?

Yes

Do you have any ethical concerns with this paper?

No

Have you any concerns about statistical analyses in this paper?

Yes

Recommendation?

Accept with minor revision (please list in comments)

Comments to the Author(s)

As per my knowledge, there are multiple things required to revise in your manuscript.

1. Addition of citations
 2. Addition of missed information about defensive behavior in honeybees like 'bee carpet' and some others.
 3. Clarification regarding microphone positioning and prevention in the colony, and recognition of acoustic signals.
 4. Correct reference writing.
 5. Calculation methods of mean signaling rate.
 6. About statistical models- did you checked the distribution of data?
 7. Filter or clean-up extraneous or noisy audio recordings, and generate spectrogram to compare bees' acoustic signatures under hornet attack and control. I would suggest to add a figure of spectrograms for this comparison.
 8. Did you examine the link between the number of predators [hornets] and production of anti-predator pipes?
 9. Include the role of soundscape indices features in recognizing the acoustic patterns of bees that emit volatiles/chemical compounds on exposure of nasonov glands.
 10. Clarification on some questions like; a] Does the predator also produce sound before or during the fight against the bees? In what frequency?
b]- Does the strength of the colony effect which acoustic frequencies are produced? Or the number of attacking organisms?
 11. Authors should add significant marks in all figures [2,4,5,6] homogenously.
 12. Table 1 style must be drawn according to the journal's instruction [three line table]. For this, authors can check the style from any published article of this journal.
- To greatly revise this manuscript, have a look on my detailed review comments (see Appendix A).

Review form: Reviewer 2

Is the manuscript scientifically sound in its present form?

Yes

Are the interpretations and conclusions justified by the results?

Yes

Is the language acceptable?

Yes

Do you have any ethical concerns with this paper?

No

Have you any concerns about statistical analyses in this paper?

No

Recommendation?

Accept with minor revision (please list in comments)

Comments to the Author(s)

Giant hornet (*Vespa soror*) attacks trigger frenetic antipredator signalling in honey bee (*Apis cerana*) colonies, appears to be an interesting scientific manuscript, of original research on the honeybees' acoustical communication during pray-predator interaction. It can advance scientific knowledge on this field. It includes a relatively extensive load of data and conclusions are supported by results and analysis. In general, the manuscript is well written; however, there are few areas that require some improvement.

More specific:

-The length of the manuscript is unnecessarily extensive and should be reduced. Especially at the sections of Introduction and Discussion. You must reduce the length and keep it focused just on the subject as described by its title.

-Lines 152-153. Overheating and asphyxiating are two different ways of bees killing the hornets without stinging. It's better if you separate the cases and the references at this sentence

-Line 215. Why have you decided to use this range of frequencies covered by this type of microphone? In literature there are experiments applied with microphones covering a wider range of frequencies when recording acoustical communication of honeybees (and also during honeybees-hornets interactions)

-Lines 223-225. This sentence is unclear though it seems important. I am not sure what do the authors mean. Recording other sounds outside the hive does not mean that they record all the bees' sound inside the hive (many differences especially in high frequencies between recorded bees and a variety of sounds that might be recorded from the background).

-General question regarding methodology: Have you applied any "control" recordings (i.e. recordings in hives without bees) to check the spectrum of background sounds (and noise) and "mask" your recording by removing them? If yes you must provide them in the manuscript.

-Line 610. Indeed, the function of antipredator pipes require confirmation. Have you tried to reproduce the sounds artificially, at the absence of hornets' attacks, and observe if they can result in any behavioural response by the nest mates? I understand that this is not the objective of the study but I am just wondering since you mention the need of playback experiments at line 661

-Lines 621-623. To my knowledge, Cyprian honeybees engulf *Vespa orientalis* in order to induce asphyxia (not the case of heat-balling). Please, have a look at *Current Biology* 17, R795-796

Decision letter (RSOS-211215.R0)

Dear Dr Mattila

On behalf of the Editors, we are pleased to inform you that your Manuscript RSOS-211215 "Giant hornet (*Vespa soror*) attacks trigger frenetic antipredator signalling in honey bee (*Apis cerana*) colonies" has been accepted for publication in Royal Society Open Science subject to minor

revision in accordance with the referees' reports. Please find the referees' comments along with any feedback from the Editors below my signature.

Please submit your revised manuscript and required files (see below) no later than 7 days from today's (ie 29-Sep-2021) date. Note: the ScholarOne system will 'lock' if submission of the revision is attempted 7 or more days after the deadline. If you do not think you will be able to meet this deadline please contact the editorial office immediately.

on behalf of Dr David Wilson (Associate Editor) and Kevin Padian (Subject Editor)
openscience@royalsociety.org

Associate Editor Comments to Author (Dr David Wilson):

Associate Editor: 1

Comments to the Author:

The study describes vibroacoustic signalling behaviour of Asian honeybees during natural encounters with (1) dangerous giant hornets that threaten the entire colony and (2) with less-dangerous smaller hornets that threaten individuals. A third treatment exposes bees to van Der Vecht gland excretions of giant hornets, and two control treatments document signalling behaviour in the absence of predator cues. Results show that bees produce multiple types of vibroacoustic signals, possibly in conjunction with chemical signals, and that the rate of signalling increases with increasing threat. Perhaps most significant is the discovery of a previously undocumented signal that is structurally distinct from other signals and which is produced specifically and immediately in response to the presence of giant hornets. The study makes a strong argument that these eusocial insects have evolved a complex referential alarm communication system, though it correctly acknowledges that playback studies documenting receiver responses in the absence of predator cues will be needed to fully understand signal function.

Both reviewers believe the manuscript is interesting and significant, and both make several suggestions for minor revisions that will make it even stronger. The writing and presentation of the manuscript are excellent, but I agree with reviewer 2 that it is long. Where possible, please reduce the length, especially in the introduction and discussion. The first half of the third

paragraph of the discussion, for example, is largely redundant with the results and could be removed. Both reviewers and I would like some clarification about defensive behaviours: reviewer 2 - differentiating between overheating and asphyxiating, reviewer 1 - describing 'bee carpet' behaviour, myself - explaining how body shaking deters predators. Both reviewers would like some clarification about the acoustical methods; I would also like to see the microphone model, recorder settings (sampling rate, bit rate, and file format), spectrogram settings (% overlap and windowing function - e.g., Hamming), a definition of 'frequency modulation' (it varies among researchers), and clarification of how events on the audio recordings were synchronized with events on the video given that they were recorded on separate devices with different (possibly drifting) clocks (L325). Reviewer 1 would like some clarification about the statistical methods, especially how mean signalling rates were calculated and whether the data met the assumptions of the parametric statistical models (I note that some variables are strongly skewed - see fig. 3); I am also curious whether colony size (e.g., number of bees) was known and whether that should be incorporated as a covariate in the models (since more bees presumably mean more signals) to better account for variance.

Minor edits: L93 - change to 'a class of'; L110 - missing period; L207 - change to 'in a vial'; L406 - there is no section '2b'; Table 1 - should "B" be "b" in rows 3 and 4?; Fig 1 caption - change 'strength' to 'significance', as 'strength' implies effect size

Reviewer comments to Author:

Reviewer: 1

Comments to the Author(s)

As per my knowledge, there are multiple things required to revise in your manuscript.

1. Addition of citations
 2. Addition of missed information about defensive behavior in honeybees like 'bee carpet' and some others.
 3. Clarification regarding microphone positioning and prevention in the colony, and recognition of acoustic signals.
 4. Correct reference writing.
 5. Calculation methods of mean signaling rate.
 6. About statistical models- did you checked the distribution of data?
 7. Filter or clean-up extraneous or noisy audio recordings, and generate spectrogram to compare bees' acoustic signatures under hornet attack and control. I would suggest to add a figure of spectrograms for this comparison.
 8. Did you examine the link between the number of predators [hornets] and production of anti-predator pipes?
 9. Include the role of soundscape indices features in recognizing the acoustic patterns of bees that emit volatiles/chemical compounds on exposure of nasonov glands.
 10. Clarification on some questions like; a] Does the predator also produce sound before or during the fight against the bees? In what frequency?
b]- Does the strength of the colony effect which acoustic frequencies are produced? Or the number of attacking organisms?
 11. Authors should add significant marks in all figures [2,4,5,6] homogenously.
 12. Table 1 style must be drawn according to the journal's instruction [three line table]. For this, authors can check the style from any published article of this journal.
- To greatly revise this manuscript, have a look on my detailed review comments from attached "Review Comments Form".

Reviewer: 2

Comments to the Author(s)

Giant hornet (*Vespa soror*) attacks trigger frenetic antipredator signalling in honey bee (*Apis cerana*) colonies, appears to be an interesting scientific manuscript, of original research on the honeybees' acoustical communication during prey-predator interaction. It can advance scientific knowledge on this field. It includes a relatively extensive load of data and conclusions are supported by results and analysis. In general, the manuscript is well written; however, there are few areas that require some improvement.

More specific:

-The length of the manuscript is unnecessarily extensive and should be reduced. Especially at the sections of Introduction and Discussion. You must reduce the length and keep it focused just on the subject as described by its title.

-Lines 152-153. Overheating and asphyxiating are two different ways of bees killing the hornets without stinging. It's better if you separate the cases and the references at this sentence

-Line 215. Why have you decided to use this range of frequencies covered by this type of microphone? In literature there are experiments applied with microphones covering a wider range of frequencies when recording acoustical communication of honeybees (and also during honeybees-hornets interactions)

-Lines 223-225. This sentence is unclear though it seems important. I am not sure what do the authors mean. Recording other sounds outside the hive does not mean that they record all the bees' sound inside the hive (many differences especially in high frequencies between recorded bees and a variety of sounds that might be recorded from the background).

-General question regarding methodology: Have you applied any "control" recordings (i.e. recordings in hives without bees) to check the spectrum of background sounds (and noise) and "mask" your recording by removing them? If yes you must provide them in the manuscript.

-Line 610. Indeed, the function of antipredator pipes require confirmation. Have you tried to reproduce the sounds artificially, at the absence of hornets' attacks, and observe if they can result in any behavioural response by the nest mates? I understand that this is not the objective of the study but I am just wondering since you mention the need of playback experiments at line 661

-Lines 621-623. To my knowledge, Cyprian honeybees engulf *Vespa orientalis* in order to induce asphyxia (not the case of heat-balling). Please, have a look at *Current Biology* 17, R795-796

===PREPARING YOUR MANUSCRIPT===

===PREPARING YOUR REVISION IN SCHOLARONE===

Author's Response to Decision Letter for (RSOS-211215.R0)

See Appendix B.

Decision letter (RSOS-211215.R1)

Dear Dr Mattila,

I am pleased to inform you that your manuscript entitled "Giant hornet (*Vespa soror*) attacks trigger frenetic antipredator signalling in honey bee (*Apis cerana*) colonies" is now accepted for publication in Royal Society Open Science.

You can expect to receive a proof of your article in the near future. Please contact the editorial office (openscience@royalsociety.org) and the production office (openscience_proofs@royalsociety.org) to let us know if you are likely to be away from e-mail contact -- if you are going to be away, please nominate a co-author (if available) to manage the proofing process, and ensure they are copied into your email to the journal. Due to rapid

publication and an extremely tight schedule, if comments are not received, your paper may experience a delay in publication.

on behalf of Dr David Wilson (Associate Editor) and Kevin Padian (Subject Editor)
openscience@royalsociety.org

Appendix A

Review Comments for Authors

Line numbers	Reviewers' comments
107	Here, also describe acoustic emissions of 6 kHz frequency produced by Apis mellifera while confronting with V. orientalis [Papachristoforou et al., 2008].
157	Authors have missed one of the important defensive behaviors against predators in honeybees, and that behavior is designated as "Bee-carpet". I would suggest adding a couple of sentences about this important behavior. For detailed defensive behavior in honeybees read a paper by Sharif et al. [2020] with title "Insect pests, parasitoids, and predators; Can they degrade the sociality of a honeybee colony, and be assessed via acoustically monitored systems?"
189	From the methodology section [2.2], I have the following questions to ask; 1- As you put the microphone close to the cluster of bees then how did you prevent the microphone from propolization? 2- How did you differentiate the acoustic signals during the attack? Meaning, which acoustic signal is from bees and which from hornet [predator]?
229-230	The reference "Cornell Lab of Ornithology, v. 1.5, www.birds.cornell.edu/raven)", I think that common reference for that software is "[Bioacoustics Research Program 2014]". Please check. Bioacoustics Research Program [2014] Raven Pro; Interactive sound analysis software. Version 1.5. Computer software. The cornel lab of ornithology, Ithaca, NY. Available from https://www.birds.cornell.edu/raven
247-248	Briefly, describe how did you calculate the mean signaling rate/minute? And citation?
303-305	How do you know that one-way and two-way ANOVAs are the best models for your data? Did you check the distribution of data?
339	Why didn't you filter or clean up extraneous or noisy audio recordings? It is better to filter a part of audio recordings by using Audition or Raven software. After that generate a spectrogram to compare bees' acoustic signatures under hornet attack and control. I would suggest adding a figure of spectrograms for this comparison.

489	Did you examine the link between the number of predators [hornets] and the production of anti-predator pipes? If you have data then it would be better to include it in the result.
767	Nasonov glands in workers release various volatile compounds or chemicals which help the colony in locating home, food, and water resources, and inform about swarming and intruders/danger [hornets, wasps]. If we could know how honeybee colonies respond to these volatile compounds or chemicals then we possibly understand the purpose of the colony's acoustic emissions. If colony soundscape signatures produced in response to volatile compounds or other chemicals through the exposure of nasonov gland are analyzed by employing a feature called ' Soundscape Indices ' then we would be able to understand the messages [swarming, danger, food, and water resources] of volatiles/chemicals producing colony workers. I would suggest citing the role of 'soundscape indices' against different chemicals in the discussion section as mentioned in an article by Sharif et al. [2020] entitled; ' Soundscape indices: New features for classifying beehive audio samples '.
790	As mentioned in a review article by Sharif et al. [2020], did you notice the following things in your experiment? 1-Does the predator also produce sound before or during the fight against the bees? In what frequency? 2- Does the strength of the colony affect which acoustic frequencies are produced? Or the number of attacking organisms? If you have some data regarding the above-mentioned questions, it would be great to add otherwise cite these questions for future exploration.
855, 872, 885,899	Authors should add significant marks in all figures [2,4,5,6] homogenously.
838	Table 1 style must be drawn according to the journal's instruction [three line table]. For this, authors can check the style from any published article of this journal.

Appendix B

RESPONSE TO THE REVIEWS

We thank the Editor and both Reviewers for their thoughtful comments about our manuscript. We have tried in good faith to address all of their ideas, concerns, and suggestions. As the Editor points out, many of the comments were similar, which provided strong guidance for revisions that have improved the clarity of our paper. Our specific responses are outlined below, point by point.

Associate Editor Comments to Author (Dr. David Wilson):

The study describes vibroacoustic signalling behaviour of Asian honeybees during natural encounters with (1) dangerous giant hornets that threaten the entire colony and (2) with less-dangerous smaller hornets that threaten individuals. A third treatment exposes bees to van Der Vecht gland excretions of giant hornets, and two control treatments document signalling behaviour in the absence of predator cues. Results show that bees produce multiple types of vibroacoustic signals, possibly in conjunction with chemical signals, and that the rate of signalling increases with increasing threat. Perhaps most significant is the discovery of a previously undocumented signal that is structurally distinct from other signals and which is produced specifically and immediately in response to the presence of giant hornets. The study makes a strong argument that these eusocial insects have evolved a complex referential alarm communication system, though it correctly acknowledges that playback studies documenting receiver responses in the absence of predator cues will be needed to fully understand signal function.

Both reviewers believe the manuscript is interesting and significant, and both make several suggestions for minor revisions that will make it even stronger. The writing and presentation of the manuscript are excellent, but I agree with reviewer 2 that it is long. Where possible, please reduce the length, especially in the introduction and discussion. The first half of the third paragraph of the discussion, for example, is largely redundant with the results and could be removed.

We strongly edited the Introduction and Discussion to keep the direction of those sections focused. In the Introduction, we removed a half page of writing (~200 words), including several references, although we had to add 1.5 sentences about specific defenses in response to questions from the Editor and both Reviewers (see next response). The Results section was trimmed by ~150 words. Major cuts were made to the Discussion, which was reduced in length by about one third (~1200 words).

Both reviewers and I would like some clarification about defensive behaviours: reviewer 2 - differentiating between overheating and asphyxiating, reviewer 1 - describing 'bee carpet' behaviour, myself - explaining how body shaking deters predators.

*To address these concerns, we elaborated on our explanation of *A. cerana* defenses against hornets in the Introduction in the following ways:*

The combination of papers we cite about bee balling (refs 89,94-96 in the Introduction) together tell the story that A. cerana kills hornets in “heat” balls both by overheating them and depriving them of oxygen, which is why we say a heat ball kills a hornet by “simultaneously overheating and asphyxiating it [89,94-96]”. Refs 89+94 highlight the importance of heat in A. cerana, but then ref 95 showed it wasn’t lethal without simultaneous oxygen deprivation. Ref 96 shows that asphyxiation and heating both occur when A. mellifera cypria balls Oriental hornets. Ref 95 is Sugahara and Sakamoto (2009) Naturwissenschaften (paper title: Heat and carbon dioxide generated by honeybees jointly act to kill hornets) and ref 96 is Papachristoforou et al. (2007) Current Biology, which states “Although temperature is not sufficient to kill the Oriental hornet, it does contribute to its death when combined with a malfunction of the respiratory system.” In the interest of briefly summarizing a behavior that is not the focus of the paper, we feel this sentence provides what is needed for the Introduction about how balling works. To avoid overemphasizing heating, we edited the term “heat ball” to “bee ball” or simply “ball” throughout the manuscript.

“Bee carpets” are a term used in the literature to describe the aggregations that A. mellifera workers make at entrances prior to bee balling. While the term “bee carpet” has not been applied to Asian bees (that we are aware), we fully agree that we need to mention these entrance aggregations in the Introduction, especially because we documented changes in bee numbers on hive fronts in our study. We added this statement about A. cerana: ““They often aggregate at the nest entrance as a first step [70,88,89], referred to a “bee carpets” in A. mellifera [90-93]” before describing the defenses that follow these entrance aggregations (ie. bee balling, dung spotting, body shaking).

Finally, we elaborated on the means by which body shaking is thought to work against predators: “Groups of workers also perform coordinated body shaking in response to hornets, a visually intimidating display that deters attackers from approaching the nest [77,94–97].”

Both reviewers would like some clarification about the acoustical methods; I would also like to see the microphone model, recorder settings (sampling rate, bit rate, and file format), spectrogram settings (% overlap and windowing function - e.g., Hamming), a definition of 'frequency modulation' (it varies among researchers), and clarification of how events on the audio recordings were synchronized with events on the video given that they were recorded on separate devices with different (possibly drifting) clocks (L325).

We added these details to the Methods, including info on the recorder settings (16 bit rate, 44.1 kHz sampling rate, 50-20,000 Hz frequency range, mp3 files made by the recorder that were converted to wav files by Raven), microphone source (flat-frequency lavalier, frequency range 50-15,000 Hz, Mediamart Joint Stock Company, Hanoi Vietnam, no model given), spectrogram settings (brightness = 65%; contrast = 75%; spectrogram window size = 1200 points, 21.5 Hz resolution; smoothing = on; colour map = hot; 50% overlap; Hann window function).

We defined “frequency modulation” according to Schlegel et al. 2012 in the text as “change in frequency over the duration of a signal [64]”.

We also described our method for synchronizing the audio and video recordings: “Clocks for videos were synchronized with corresponding audio recordings using a verbal mark that was announced at the start of both recordings.”

Reviewer 1 would like some clarification about the statistical methods, especially how mean signalling rates were calculated and whether the data met the assumptions of the parametric statistical models (I note that some variables are strongly skewed-see fig. 3);

*We expanded our explanation in the Methods about this calculation: “Using this final dataset of categorized signals, we calculated the number of each signal type that was produced per minute in each colony replicate. These values **were averaged across minutes** to determine mean rates of signalling per minute for each colony (for each signal type separately and for all signal types combined).” We think the request for a citation was given with the thought that the calculation was complicated, but it is straightforward and doesn’t require a citation.*

We tested the normality and homogeneity of variances of datasets that were subjected to parametric tests (ANOVAs and t-tests) and applied log transformations as necessary. The transformations either met the assumption of normality or improved normality; variance homogeneity was the norm. When homogeneity of variances is maintained, recent modeling shows that ANOVAs are highly robust to deviations from normality and maintain acceptable Type I error rates, according to Blanca et al. (2017), a statistical study that is highly cited by researchers (318 other studies to date). We cite it as well. We appreciate the push to more closely examine the distribution of our data. This revision required minor adjustments to reported ANOVA values whenever transformations were applied, but it did not change any of our conclusions. Original and transformed data are provided in our data file.

I am also curious whether colony size (e.g., number of bees) was known and whether that should be incorporated as a covariate in the models (since more bees presumably mean more signals) to better account for variance.

Unfortunately, we did not collect data on colony size, so we can’t incorporate it into our statistical approach. We have to interpret our data with the experimental unit as the “colony”, without knowing how it varied in size across replicates.

Minor edits:

L93 - change to 'a class of'

Fixed it, thanks!

L110 - missing period

Added.

L207 - change to 'in a vial'

Fixed it, thank you!

L406 - there is no section '2b'

Thank you, corrected to "section 3.2.4."

Table 1 - should "B" be "b" in rows 3 and 4

Yes, thank you! We fixed it.

Fig 1 caption - change 'strength' to 'significance', as 'strength' implies effect size

Good point, we fixed this mistake throughout all relevant fig captions.

Reviewer 1

As per my knowledge, there are multiple things required to revise in your manuscript.

1. Addition of citations:

Line 107 Here, also describe acoustic emissions of 6 kHz frequency produced by *Apis mellifera* while confronting with *V. orientalis* [Papachristoforou et al., 2008].

*We cited this reference in the Discussion (ref 111), but we didn't add it to line 107 because that part of the Introduction is about stop signals. Stop signals are short and linked to head butting, which is nothing like the sound made by Cyprian bees in response to hornets (see Fig 1 in ref 111). In fact, it is much closer to the antipredator pipes that our paper describes, which is why we bring it up in the Discussion instead, after we have given our documentation of a similar sound in *A. cerana* colonies. ("The signal traits of antipredator pipes also closely match those made by *A. mellifera* cypria when they try to ball *Vespa orientalis* [111])"*

The reference "Cornell Lab of Ornithology, v. 1.5, www.birds.cornell.edu/raven)", I think the common reference for that software is "[Bioacoustics Research Program 2014]". Please check. Bioacoustics Research Program [2014] Raven Pro; Interactive sound analysis software. Version 1.5. Computer software. The cornel lab of ornithology, Ithaca, NY. Available from <https://www.birds.cornell.edu/raven>

Thank you for noting this point. We looked on the Raven Pro website, where they request that citations for their software be given as follows:

K. Lisa Yang Center for Conservation Bioacoustics. (2014). Raven Pro: Interactive Sound Analysis Software (Version 1.5) [Computer software]. Ithaca, NY: The Cornell Lab of Ornithology. Available from <http://ravensoundsoftware.com/>.

We added this citation to the references.

Line 767 Nasonov glands in workers release various volatile compounds or chemicals which help the colony in locating home, food, and water resources, and inform about swarming and intruders/danger [hornets, wasps]. If we could know how honeybee colonies respond to these volatile compounds or chemicals then we possibly understand the purpose of the colony's acoustic emissions. If colony soundscape signatures produced in response to volatile compounds or other chemicals through the exposure of nasonov gland are analyzed by employing a feature called 'Soundscape Indices' then we would be able to understand the messages [swarming, danger, food, and water resources] of volatiles/chemicals producing colony workers. I would suggest citing the role of 'soundscape indices' against different chemicals in the discussion section as mentioned in an article by Sharif et al. [2020] entitled; 'Soundscape indices: New features for classifying beehive audio samples'.

The reviewer brings up an interesting idea, but one that is beyond the scope of what we can speculate about in our Discussion. We measured sounds, but we did not measure chemical emissions by alarmed workers. We noted that they exposed their Nasonov glands at entrances, and we strongly believe that future work should explore the potential multimodal nature of the vibroacoustic and chemical alarm signals released by workers when hornets attack their nest. Presently, we don't know how these chemical signals travel within colonies or which colony members they might affect. Thus, we feel it stretches the limits of our study to make guesses about how chemical signals alone could influence colony-level signalling (in the absence of hornet-related stimuli). Moreover, we argue that one of the main drivers of the signals we recorded were the hornets themselves. It would be hard to disentangle the hornets as a cause of signaling from the effect of worker-produced chemicals as a cause of signaling. It sounds like the reviewer is advocating for the latter, similar to the Sharif et al. 2020 paper that showed that exposing colonies to noxious chemicals causes them to make signature sounds, according to Sharif et al.'s 'soundscape indices'. However, focusing only on chemicals would counter one of our main conclusions for this study. We think that giant hornets have to be present to stimulate workers to generate these soundscapes.

However, we added Sharif et al. (2020) as a citation in the Introduction, when we mentioned the utility of acoustic monitoring in complex environments in which sound is a well-used modality and visual observation is challenging.

2. Addition of missed information about defensive behavior in honeybees like 'bee carpet' and some others. Line157 Authors have missed one of the important defensive behaviors against predators in honeybees, and that behavior is designated as "Bee-carpet". I would suggest adding a couple of sentences about this important behavior. For detailed defensive behavior in honeybees read a paper by Sharif et al. [2020] with title "Insect pests, parasitoids, and

predators; Can they degrade the sociality of a honeybee colony, and be assessed via acoustically monitored systems?"

*As described above, we added information about aggregations of workers at colony entrances (called bee carpets in *A. mellifera*), which occurs as a first step in bee balling, dung spotting, or body shaking.*

3. Clarification regarding microphone positioning and prevention in the colony, and recognition of acoustic signals. Line 189 From the methodology section [2.2], I have the following questions to ask; 1- As you put the microphone close to the cluster of bees then how did you prevent the microphone from propolization?

A. cerana does not make propolis, so there was no risk of having the microphones propolized by the bees.

4. Correct reference writing.

We double checked all of the references to make sure that they were in the journal's style, as recommended by the editors.

5. Calculation methods of mean signaling rate. Briefly, describe how did you calculate the mean signaling rate/minute? And citation?

We addressed this question with an elaboration; see our previous response to the Editor.

6. About statistical models- did you checked the distribution of data? Line 247-248 303-305 How do you know that one-way and two-way ANOVAs are the best models for your data? Did you check the distribution of data?

We checked the normality of data and applied log transformations to either meet the normality assumption of parametric tests or improve normality; see our previous response to the Editor.

7. Filter or clean-up extraneous or noisy audio recordings, and generate spectrogram to compare bees' acoustic signatures under hornet attack and control. I would suggest to add a figure of spectrograms for this comparison. Line339 Why didn't you filter or clean up extraneous or noisy audio recordings? It is better to filter a part of audio recordings by using Audition or Raven software. After that generate a spectrogram to compare bees' acoustic signatures under hornet attack and control. I would suggest adding a figure of spectrograms for this comparison.

*Our conclusion after initial exploration of our audio recordings was that filtering was not necessary. We took this approach after consultation with personnel from Raven Pro software support. Given that we were listening to *Apis cerana-Vespa soror* soundscapes that were entirely new to us (and had not been examined before in the literature), we were concerned*

about listening to recordings that were not “complete” because they had been filtered or cleaned up. We wanted to listen to the raw audio to hear everything that was available to review. Initially, we played with some filtering, but we neither preferred to listen to altered audio nor did we need the soundscapes to be filtered to answer our questions. Because all of our Raven analyses were completed manually (no use of automation or algorithms to identify signals), we could count signals and determine their duration without filtering. The only metric we could not always discern was fundamental frequency. However, our large dataset (30K signals) made it possible to manually identify a large subset of “clean” signals for which the background noise was minimal enough that fundamental frequency could be extracted and compared across treatments (761 signals). Our early exploration of filtering did not improve our ability to discern fundamental frequencies.

We don't provide a side-by-side visual comparison of the bees' acoustic signatures in the hornet scenarios we recorded, but we do provide 1-minute wav clips from treatment groups for an audio comparison in the supplementary material (Audio S1-S4). These clips can be run through Raven to generate spectrograms (which we have tried). It would be a large figure, but we are happy to add it if the editors think it should be in the paper. Because the manuscript is already long, we chose not to include another figure with this revision.

8. Did you examine the link between the number of predators [hornets] and production of anti-predator pipes? Line 489 Did you examine the link between the number of predators [hornets] and the production of anti-predator pipes? If you have data then it would be better to include it in the result.

*Good question! We took a look at our data from Figs 8 and S4 (the replicates in which *V. soror* workers attacked colonies) and determined that there was no difference between antipredator piping rates when one hornet versus multiple hornets were present. [There was only one case when 3 hornets were present, so we collapsed 2 hornet and 3 hornet scenarios into a single “multiple hornet” group.] We added this sentence to section 3.3: “Rates of antipredator piping were similar whether there was a lone *V. soror* worker or multiple workers attacking (mean 18 ± 5 versus 20 ± 3 antipredator pipes/min, respectively; t-test: $t_{15} = 0.3$, $p = 0.74$). ” Thank you for the suggestion.*

9. Include the role of soundscape indices features in recognizing the acoustic patterns of bees that emit volatiles/chemical compounds on exposure of nasonov glands.

We referred to Sharif et al. (2020) in the Introduction, as mentioned previously.

10. Clarification on some questions like; a] Does the predator also produce sound before or during the fight against the bees? In what frequency? 2- How did you differentiate the acoustic signals during the attack? Meaning, which acoustic signal is from bees and which from hornet [predator]? Lines 229-230 790 As mentioned in a review article by Sharif et al. [2020], did you notice the following things in your experiment? 1-Does the predator also produce sound before or during the fight against the bees? In what frequency?

We have no evidence that the hornets made sounds when attacking the colonies. We didn't hear any sounds coming from them, although they are known to make audible noises when they are defending their own nests against intruders (like the bees in this study). If they made similar sounds when attacking, we likely would have heard them, similar to how the bees' signals were audible to the human ear.

We presume that all recorded signals were made by Apis cerana workers because our microphones were inside hives and hornets did not enter the hives when we were recording. Furthermore, the sounds we characterized are all previously known from Apis species, except for antipredator pipes. We are certain that antipredator pipes were produced by bees because when we ran the audio tracks through Raven Pro from the supplementary videos (of A. cerana workers making antipredator pipes in the absence of V. soror; Videos S2-S4), the pipes in the spectrograms were similar to the antipredator pipes identified from recordings made within colonies.

2- Does the strength of the colony affect which acoustic frequencies are produced? Or the number of attacking organisms? If you have some data regarding the above-mentioned questions, it would be great to add otherwise cite these questions for future Exploration.

As mentioned previously, we don't have information on colony strength, but we added information about the effect of the number of attacking V. soror workers on colony signaling.

11. Authors should add significant marks in all figures [2,4,5,6] homogenously. Line 855, 872, 885,899 Authors should add significant marks in all figures [2,4,5,6] homogenously.

We made all significance formatting homogeneous across figures.

12. Table 1 style must be drawn according to the journal's instruction [three line table]. For this, authors can check the style from any published article of this journal. Line 838.

It is our understanding that the production team puts table data into the correct format for the journal. We have provided a separate, editable Table file for this purpose, as requested by the Editor.

Reviewer 2

Giant hornet (*Vespa soror*) attacks trigger frenetic antipredator signalling in honey bee (*Apis cerana*) colonies, appears to be an interesting scientific manuscript, of original research on the honeybees' acoustical communication during prey-predator interaction. It can advance scientific knowledge on this field. It includes a relatively extensive load of data and conclusions are supported by results and analysis. In general, the manuscript is well written; however, there are few areas that require some improvement.

We are happy to read that Reviewer 2 sees scientific merit in the study.

More specific:

-The length of the manuscript is unnecessarily extensive and should be reduced. Especially at the sections of Introduction and Discussion. You must reduce the length and keep it focused just on the subject as described by its title.

We made edits throughout the manuscript, especially in the Introduction and Discussion, to shorten it where possible (as mentioned previously in response to the Editor). About ~1500 words were cut.

-Lines 152-153. Overheating and asphyxiating are two different ways of bees killing the hornets without stinging. It's better if you separate the cases and the references at this sentence

*In theory, these are two ways to kill a hornet, but the literature shows that, in practice, both occur simultaneously. Mentioned previously: See Sugahara and Sakamoto (2009) *Naturwissenschaften* (paper title: Heat and carbon dioxide generated by honeybees jointly act to kill hornets) and Papachristoforou et al. (2007) *Current Biology*, which states "Although temperature is not sufficient to kill the Oriental hornet, it does contribute to its death when combined with a malfunction of the respiratory system."*

-Line 215. Why have you decided to use this range of frequencies covered by this type of microphone? In literature there are experiments applied with microphones covering a wider range of frequencies when recording acoustical communication of honeybees (and also during honeybees-hornets interactions)

*The range of frequencies recorded in this study (0-15,000 Hz) are typical for recording vibroacoustic signals from honey bees. For example, here are three well-cited papers that recorded honey bee sounds using microphones with similar (or smaller) frequency ranges than ours: Seeley et al. (2012) in *Science*, 70-16,000 Hz; Lau and Nieh (2010) in *Apidologie*, 30-10,000 Hz; Thom et al. (2003) in *Behavioral Ecology and Sociobiology*, 20-6,000 Hz. All three papers include senior PIs who have extensive experience investigating the vibroacoustic sounds produced by honey bees.*

-Lines 223-225. This sentence is unclear though it seems important. I am not sure what do the authors mean. Recording other sounds outside the hive does not mean that they record all the bees' sound inside the hive (many differences especially in high frequencies between recorded bees and a variety of sounds that might be recorded from the background).

We agree that the statement is confusing, so we deleted it in the interest of shortening the text.

-General question regarding methodology: Have you applied any “control” recordings (i.e. recordings in hives without bees) to check the spectrum of background sounds (and noise) and “mask” your recording by removing them? If yes you must provide them in the manuscript.

We did not do this, and explained previously why we decided to extract data from unfiltered recordings.

-Line 610. Indeed, the function of antipredator pipes requires confirmation. Have you tried to reproduce the sounds artificially, in the absence of hornets’ attacks, and observe if they can result in any behavioural response by the nest mates? I understand that this is not the objective of the study but I am just wondering since you mention the need of playback experiments at line 661

We agree with Reviewer 2 that this is a necessary step to confirm the function of antipredator pipes, but we have not tried it yet. We describe playback experiments as a next step for investigation in the Discussion (“Playback experiments are necessary to resolve the response of workers to this signal.”)

-Lines 621-623. To my knowledge, Cyprian honeybees engulf *Vespa orientalis* in order to induce asphyxia (not the case of heat-balling). Please, have a look at Current Biology 17, R795-796

*As we mentioned previously, it is the case that *A. mellifera cypria* asphyxiates hornets, but bees also heat them during this defense (although not to lethal temperatures). Because all three reviewers mentioned this point, we include Papachristoforou et al. 2007 as a citation for our description of bee balling in *Apis cerana* (heating + asphyxiation), even though it reports on balling in a different species.*